# SOFT MERGING OF EXPERTS WITH ADAPTIVE ROUTING

## ABSTRACT

Neural networks that learn to route their inputs through different "expert" sub-networks provide a form of modularity that standard dense models lack. Despite their possible benefits, modular models with learned routing often underperform their parameter-matched dense counterparts as well as models that use non-learned heuristic routing strategies. In this paper, we hypothesize that these shortcomings stem from the gradient estimation techniques used to train modular models that use non-differentiable discrete routing decisions. To address this issue, we introduce **S**oft **M**erging of **E**xperts with **A**daptive **R**outing (SMEAR), which avoids discrete routing by using a single "merged" expert constructed via a weighted average of all of the experts' parameters. By routing activations through a single merged expert, SMEAR does not incur a significant increase in computational costs and enables standard gradient-based training. We empirically validate that models using SMEAR outperform models that route based on metadata or learn routing through gradient estimation. Furthermore, we provide qualitative analysis demonstrating that the experts learned via SMEAR exhibit a significant amount of specialization.

## 1 INTRODUCTION

Neural networks typically use all of their parameters to process a given input. As such, the capabilities of a model are distributed across the parameters of a model in a self-organizing way (Zeiler & Fergus, 2014; De Cao et al., 2021; Csordás et al., 2021; Bau et al., 2020; Wang et al., 2022a). Explicitly specializing different parts of a model to different capabilities can provide various benefits, including reduced interference across downstream tasks (Sanh et al., 2021; Wei et al., 2021; Zamir et al., 2018; Bao et al., 2021) or languages (Pires et al., 2019; Liu et al., 2020; Xue et al., 2020). Furthermore, dedicating specific parameters to specific capabilities enables a form of modularity where a capability can be added, removed, or modified by adding, removing, or modifying the corresponding parameters (Pfeiffer et al., 2023). Activating only a subset of the model's parameter for a given input also decouples the computational cost of a model from the number of parameters it has (Shazeer et al., 2017; Fedus et al., 2021), though we do not focus on this benefit in this paper.

*Conditional computation* techniques provide a way to build models that adaptively choose a subset of their parameters to apply to a given input. A common way to use conditional computation in this setting is to introduce specialized subnetworks called *experts* that are controlled by *routers* that decide which experts should be active. When the model is trained on diverse data, this form of conditional computation can enable modular learning by allowing experts to specialize to different types of inputs and flexibly share knowledge (Ma et al., 2019). However, because routing involves making a discrete decision as to which expert to use, the loss on the model's prediction cannot back-propagate though the routing decision to update the router. Consequently, models with conditional computation often require gradient estimation techniques for training (Clark et al., 2022; Fedus et al., 2021; Bengio et al., 2013). In practice, past work has shown that models with conditional computation do not always learn effective routing strategies. For example, Mittal et al. (2022) investigate models with a continuous router in a controlled setting and find the models do not route examples from the same group to the same experts and perform poorly compared to models with oracle routing. However, models with task- or domain-specific subnetworks (Gururangan et al., 2021; Kudugunta et al., 2021) provide evidence that it is possible to train performant models with specialized experts. As an extreme example, Roller et al. (2021) achieves results comparable to learned routing with a fixed random routing. Relatedly, Fedus et al. (2021) find the gain from scaling up parameters by 30× with a sparsely activated model is smaller than scaling up both parameters and FLOPs by 3× in a dense

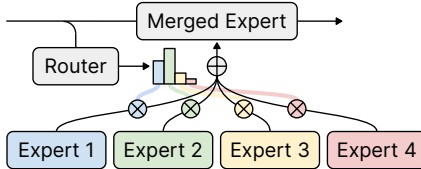

Figure 1: SMEAR uses the router's distribution to average expert parameters and routes through a single merged expert. SMEAR achieves better performance than discrete routing, can be trained with standard backpropagation, and does not incur significant additional computational costs.

model. As a possible explanation, Clark et al. (2022) study how models with conditional computation improve with scale and find a detrimental term that scales with the product of the log number of experts and active parameters.

In this work, we hypothesize that issues with conditional computation stem from issues with gradient estimation. Specifically, we focus on experimental settings where we can compare learned routing to a performant hand-designed heuristic routing scheme. We find that the gradient estimation techniques we consider often produce models that underperform heuristic routing, despite the fact that they could in principle learn a better routing strategy. To address this shortcoming, we introduce **S**oft **M**erging of **E**xperts with **A**daptive **R**outing (SMEAR), a method for training modular models with specialized experts and learned routing. SMEAR works by using the router's distribution over experts to compute a weighted average of the parameters of the individual experts. Activations are then sent through the *merged* expert, which results in a similar computational cost to discrete routing with a single expert. However, the fact that all components of SMEAR are fully differentiable enables standard gradient-based training. Empirically, we show that SMEAR significantly attains a favorable performance/cost tradeoff to 1) discrete routing solutions found via gradient estimation, 2) heuristic routing schemes, and 3) state-of-the-art baselines for learning modular models. We also qualitatively validate that the experts learned by SMEAR specialize to different types of inputs and share parameters across related tasks. Put together, our results show that SMEAR provides an effective alternative for modular models that use adaptive routing among expert subnetworks.

## 2 BACKGROUND

To provide the necessary background for our work, we first explain how sparsely activated neural networks use conditional computation, then discuss gradient estimators that enable learning discrete routing strategies. In addition, we discuss different ways to hand-design "heuristic" routing strategies as well as preexisting techniques for learning modular models that we use as baselines.

In models that use discrete routing among experts (i.e. subnetworks), experts are organized into blocks that are incorporated as an intermediate layer in a neural network. An expert routing block $B$ comprises a set of $N$ experts $\{f_1, f_2, \ldots f_N\}$ and a router $R$. Experts in the same block accept inputs and produce outputs of the same dimensionality. Given a hidden-state representation $u$, the output of the $i$-th expert with parameters $\theta_i$ is $f_i(u, \theta_i)$.

In sparsely activated models that involve discrete adaptive routing, it is not possible to train the router's parameters with standard gradient-based learning. Fortunately, gradient estimators can provide approximate gradients to the router parameters. There are a few common designs shared by models that use gradient estimators to train routers. Their router $R$ often applies a lightweight network to some intermediate hidden states $v$ in the model. The output of the lightweight routing network $R(v)$ parameterizes a discrete probability distribution over the $N$ experts. Different gradient estimators vary in how they make the routing decision from $R(v)$ and how they construct the output from the chosen expert.

**REINFORCE** Gradients can be estimated through discrete operations using reinforcement learning techniques (Schulman et al., 2015; Bengio et al., 2013). In reinforcement learning, a policy loss is used to train an agent to learn optimal actions in an environment. In this paper, we experiment with the REINFORCE algorithm which computes the policy loss as $\log(\pi)r$ where $r$ denotes the received reward for taking an action whose assigned probability is $\pi$. When applied to models that use discrete routing among experts, the goal is to train the model to choose the optimal expert to process a given input. Here, the router $R$ acts an agent that samples an expert to use according to the routing probabilities. In order to train such a router, the router's assigned probability to the sampled expert is used as $\pi$ and the negative of the model's loss is used as the reward $r$. The router is therefore trained to pick experts that maximize the reward which, in turn, minimizes the loss. The

REINFORCE estimator often suffers from high variance because of the sampling operation. This motivates the use of baselines, which reduce variance without changing the optimal solution. In our work, we follow Clark et al. (2022) and use a baseline $b$ that is generated by a small neural network with a single hidden layer that takes as input $v$ and is trained with the Huber loss. The overall loss function is then $L = -\mathbb{E}_{i \sim R(v)} \alpha \log R(v)_i (r - b) - \beta R(v) \log R(v) + \gamma L_{\text{Huber}}(r, b)$ where $\alpha$, $\beta$, and $\gamma$ are hyperparameters that correspond to policy gradient weight, policy entropy weight, and value loss weight. In practice, we approximate the expectation with a single sample. During inference, the output of the block $B$ is just $f_i(u, \theta_i)$ where $i = \arg \max R(v)$.

**Straight Through Gumbel-Softmax (ST-Gumbel)**    The Gumbel-Softmax trick (Jang et al., 2016; Maddison et al., 2016) provides a continuous differentiable approximation to sampling from a categorical distribution like the one parameterized by a router. Specifically, Gumbel noise is added to the logits of the distribution and a temperature scale is applied in the softmax operation. The expert $f_i$ with the highest assigned probability is chosen by applying an $\arg \max$ operation. In order to approximate gradients through the $\arg \max$ operation, we use the Straight-Through estimator which replaces $f_i(u, \theta_i)$ with $(1 - \text{sg}[\hat{R}(v)_i] + \hat{R}(v)_i) f_i(u, \theta_i)$ where sg stands for the stop-gradient operator. During forward pass, the multiplier for $f_i(u, \theta_i)$ becomes 1 and the multiplier receives gradients for the term $\hat{R}(v)_i$ in the backward pass. In practice, the temperature $\tau$ is gradually annealed from a high to low value so that the approximated samples are more and more similar to discrete samples. During inference, we choose an expert according to $\arg \max R(v)$.

**Top-$k$**    Shazeer et al. (2017) propose a gradient estimation scheme where the router sends the input through the $k$ experts that are assigned the highest probability. Fedus et al. (2021) later found that this router could be used effectively when $k = 1$. Specifically, the estimator selects the subnetwork with the highest probability and scales its output using its corresponding routing probability. The output of the block is therefore $R(v)_i f_i(u, \theta_i)$, where $i = \arg \max R(v)$.

**DSelect-$k$**    Hazimeh et al. (2021) proposed a differentiable approximation for the discrete Top-$k$ operation. They parameterize each selection among $N$ experts using $m$ binary variables $z_1, z_2, \ldots z_m$, produced using a learnable weight $W$ as $z = W(v)$, where $m = \log_2(N)$. The selector function $r$ takes these variables and computes $r(z)_i = \prod_{j \in B(i-1)} z_j \prod_{j \in 1, \ldots, m \setminus B(i-1)} (1 - z_j)$, where $i \in 1, \ldots, N$ and $B(l)$ returns the non-zero indices in the binary representation of the integer $l$. A differentiable step function $S$ based on a cubic polynomial is used for the $z$ variables. Finally, each selector operation happens $k$ times to get Top-$k$ selection and an additional learnable parameter $G$ provides a probability distribution over these $k$ selections using the softmax fuction. In addition, entropy regularization is applied to the output of the selector function $r$, ensuring that it results in one-hot selection during inference. In our work, we consider $k = 1$ to maintain a computational cost similar to other baselines.

As a point of comparison for techniques that learn adaptive routing, we experiment with three baseline routing strategies that do not require a trained router.

**Tag Routing**    If we have prior knowledge about the data that a model will be applied to, we can hand-design a heuristic routing strategy for choosing which expert to use for a given example based on data properties. Tag routing takes advantage of "tags" associated with a given example (such as its domain or task) and associates each expert in an expert routing block with a particular tag. In this work, we assume each example has a single tag and route each example to its tag's expert.

**Hash Routing**    Roller et al. (2021) propose hash routing, which uses a fixed hashing function to determine which expert to use for a given example. Specifically, each example is assigned a random expert choice in each expert routing block which is used consistently over the course of training and inference. This approach disregards any shared characteristics across examples.

**Single-Expert**    As an additional baseline, we consider models where all inputs are routed to a single expert in each routing block. To provide a fair comparison to models with $N$ experts per block on the basis of both computational cost or parameter count, we consider models with a single expert with either the same number (compute-matched, referred to as "$1\times$ compute") or $N\times$ (parameter-matched, referred to as "$1\times$ parameters") as many parameters as a single expert.

Beyond the simple baseline discussed above, we consider three recently proposed methods that aim to learn modular models.

**Adamix**    Adamix (Wang et al., 2022b) uses random routing for each example during training and adds a consistency loss to encourage experts to share information and discourage divergence. During inference, the parameters of all experts are averaged together to form a single expert and no adaptive routing is used.

**Latent Skills**    Latent Skills (Ponti et al., 2022) assumes that the task for each example is known and trains a task-skill matrix that specifies which experts are active for a given task. The binary task-skill matrix is fixed and learned via the Gumbel-Sigmoid trick Maddison et al. (2016). During inference, a merged expert is formed for each task by averaging the parameters of the skill experts weighted according to the task-skill matrix.

**Soft MoE**    Puigcerver et al. (2023) recently proposed Soft MoE, which assigns "slots" to each expert and passes a weighted average of input tokens into each slot. All operations in Soft MoE method are differentiable, avoiding the need for the gradient estimation. We consider Soft MoE with a single slot per expert to ensure fair comparison by having computational cost equivalent to other discrete routing baselines.

## 3    SOFT MERGING OF EXPERTS WITH ADAPTIVE ROUTING

As we will later show in section 4, the gradient estimation techniques used to train models with discrete routing often fail to produce performant routing strategies. Our goal in this work is therefore to explore whether it is possible to train models with adaptive routing among experts without resorting to gradient estimation. Specifically, we aim to achieve better performance by designing an expert and router architecture that facilitates standard end-to-end gradient-based training but does not increase computational costs.

**Ensemble Routing**    One simple idea would be to pass the input of a given expert routing block through *every* expert, and then compute an average of the experts' outputs weighted according the router's distribution, i.e. exactly computing $\mathbb{E}_{i \sim R(v)} f_i(u, \theta_i)$. We refer to this approach as an *ensemble* routing strategy since it corresponds to using the ensemble prediction of the experts. Since the operations involved in computing the average are all differentiable, using an ensemble routing strategy would allow for exact computation of gradients and end-to-end-learning. Unfortunately, such an approach would incur a significant increase in computational costs because it requires computing the output of every expert rather than a single expert.

**Merging Experts**    To explore an alternative fully-differentiable expert routing block, we take inspiration from recent work on *merging* models (Matena & Raffel, 2021; Wortsman et al., 2022b;c; Choshen et al., 2022b; Don-Yehiya et al., 2022; McMahan et al., 2017). These works have shown that averaging the parameters of models that share a common architecture can often produce an aggregate model that shares the capabilities of the individual models. Notably, Wortsman et al. (2022b); Matena & Raffel (2021) found that averaging the weights of multiple fine-tuned models produced a single model that performs comparably to an ensemble of the models. In addition, both Adamix (Wang et al., 2022b) and Latent Skills (Ponti et al., 2022) include steps that involve averaging expert parameters, though neither of these methods learn an adaptive per-example routing strategy. Motivated by these findings, we propose **S**oft **M**erging of **E**xperts with **A**daptive **R**outing (SMEAR), which constructs a single merged expert whose parameters are computed as the weighted average of the experts within a routing block. Each expert's weight is set according to the corresponding routing probability generated by the router. In SMEAR, the input to the routing block is fed into the merged expert and the merged expert's output is used as the output of the block. By averaging parameters, SMEAR implicitly assumes that all experts in the routing block share an identical architecture (thereby inducing a natural one-to-one mapping between parameters in each expert). To the best of our knowledge, all past works focused on routing among experts use experts with a common architecture, so we do not see this assumption as a major limitation.

More explicitly, we define SMEAR as computing the output of an expert routing block using a merged expert computed as $\bar{f}(u, \sum_i R(v)_i \theta_i)$. The merged expert shares the same architecture with the

individual experts $f_i$. Notably, the input of the routing block is only ever processed by $\bar{f}$; activations are never fed to any of the individual experts. To break symmetry, all experts are randomly initialized with different parameter values. Importantly, all operations in SMEAR are fully differentiable, enabling standard gradient-based end-to-end learning. In addition, SMEAR retains the ability to learn an adaptive routing strategy that can route different examples to different experts without relying on hand-specified tags (as in Latent Skills and tag-based routing). We will later show qualitatively that this leads to meaningful specialization of different experts in real-world experiments.

**Computational Costs**   Importantly, SMEAR only ever computes the output of a single expert, suggesting that SMEAR's computational cost could be comparable to single-expert discrete routing and significantly lower than ensemble routing. However, the averaging operation in SMEAR incurs a nontrivial computational cost. To quantify this cost, we focus on the common expert architecture comprising a dense layer that projects from $d$-dimensional activations to an $m$-dimensional vector followed by a nonlinearity and an additional dense layer projecting from $m$ dimensions back to $d$. For simplicity, we ignore the (relatively minor) cost of the nonlinearity. We assume the input is a length-$L$ sequence of activations with size $L \times d$. In this case, computing the output of the merged experts incurs a computational cost of approximately $L \times 4 \times d \times m$ FLOPs and ensemble routing with $N$ experts would require $N \times L \times 4 \times d \times m$ FLOPs. SMEAR additionally must average together the parameters of $N$ experts, which costs an additional $N \times 2 \times d \times m$ FLOPs. Some past work on models with discrete routing has the router choose a different expert for each entry in the input sequence of activations (e.g. Fedus et al., 2021; Lewis et al., 2021; Roller et al., 2021). This would require computing the expert average $L$ times, which would make the cost of SMEAR similar to that of ensemble routing. We therefore focus on settings where models make a *single* routing choice for an entire input example (e.g. Gururangan et al., 2021; Kudugunta et al., 2021; Ye et al., 2022). This results in a total cost of approximately $(L \times 4 + N \times 2) \times d \times m$ for SMEAR. Consequently, as long as $L \times 4 \gg N \times 2$, SMEAR and discrete routing have roughly the same computational costs. Given that $L$ is on the order of hundreds or thousands of tokens for text-based tasks and on the order of thousands for vision tasks, $L \times 4$ will be much larger than $N \times 2$ as long as there is a modest number of experts. In our experiments within the T5-GLUE setting, where $L = 128$ and $N = 8$, this results in a minimal runtime difference. Furthermore, we would expect SMEAR to be approximately $\frac{N \times L}{N + L}$ times cheaper than ensemble routing. More concretely, we will later experimentally validate that the wall-clock time required to process an example with SMEAR in real-world experiments is roughly the same as using discrete routing and significant faster than ensemble routing.

## 4   EXPERIMENTS

In order to thoroughly evaluate the effectiveness of SMEAR, we perform experiments in two real-world settings that differ in model architecture and modality. We are particularly interested in whether a given approach for learning routing outperforms the heuristic routing strategies described in section 2. As such, we focus on experimental settings where a performant "tag routing" baseline can be designed, i.e. where we have oracle access to metadata that can be used to appropriately route examples. Specifically, we experiment with fine-tuning T5.1.1 Base (Raffel et al., 2020) on datasets from GLUE (Wang et al., 2018) (referred to as T5-GLUE) and fine-tuning a ResNet18 (He et al., 2016) on DomainNet (Peng et al., 2019) (ResNet-DomainNet). In these settings, we add experts to an existing pre-trained backbone in the same way that Adapters are used for parameter-efficient fine-tuning Houlsby et al. (2019). While past work has also considered using discrete routing among experts to train large-scale models from scratch (Fedus et al., 2021; Shazeer et al., 2017), we focus on modular fine-tuned models in this work and leave large-scale experiments for future work.

**T5-GLUE**   In this setting, we focus on training a T5 model (Raffel et al., 2020) on the GLUE meta-benchmark (Wang et al., 2018) for natural language understanding. We provide background on the GLUE dataset and the example format we use in appendix B.1. We follow the approach of Mahabadi et al. (2021) for splitting each GLUE dataset into train, eval, and test splits. Past work has demonstrated improved performance on RTE by co-training with MNLI (Phang et al., 2018; Devlin et al., 2018; Pruksachatkun et al., 2020; Vu et al., 2020; Choshen et al., 2022a), and we congruously found that sharing an expert between RTE and MNLI produced a stronger tag routing strategy. In the interest of making our baselines as strong as possible, we use this improved tag routing scheme in all experiments. We use the pretrained T5.1.1 Base model as the backbone and adapt the model in a

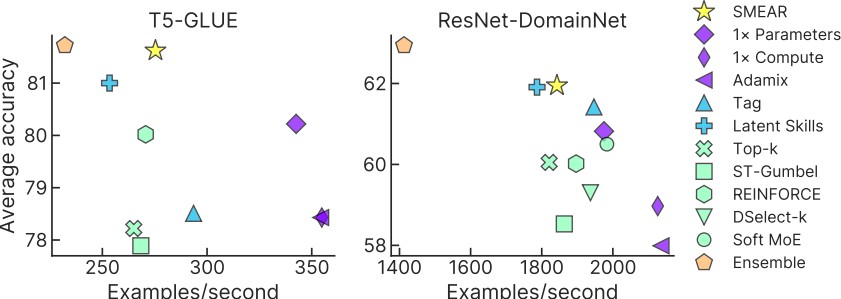

Figure 2: Average accuracy and inference speed (in examples processed per second) for models using different routing approaches on our T5-GLUE and ResNet-DomainNet settings. Routing approaches are grouped by color; groups are (in order of the legend) our method (SMEAR), methods that do not use adaptive routing (1× compute, 1× parameters, Adamix, and Hash), methods that make use of metadata (Tag and Latent Skills), methods that learn adaptive routing (Top-$k$, ST-Gumbel, REINFORCE, DSelect-$k$, and Soft MoE), and methods that ensemble expert outputs (Ensemble). We omit Hash routing from the plots because its poor performance (66.9% on T5-GLUE and 52.4% on ResNet-DomainNet) hampers readability. Exact numerical results for all methods and standard deviation across five runs are provided in appendix F.

way similar to adding adapters (Houlsby et al., 2019) for a single task, i.e. we keep all pretrained parameters frozen except for layer normalization parameters and insert expert routing blocks after self-attention, feed-forward and cross-attention modules. The T5 1.1 Base model has 12 Transformer layers in both the encoder and decoder, resulting in a total of $12 \times 2 = 24$ blocks in the encoder and $12 \times 3 = 36$ blocks in the decoder, or 60 expert routing blocks in total. In each block, we introduce eight experts (one for each dataset in GLUE). The router architecture is simply a linear classifier, i.e. a linear projection layer consisting of a weight matrix of shape $d$ x $N$, where $d$ is the model dimension and $N$ is the number of experts in the MoE layer, followed by a softmax nonlinearity. To help avoid saturating the softmax nonlinearity, we apply layer normalization both to the input of the router as well as the rows of the linear layer. In the encoder, the router takes as input the preceding hidden states, which are averaged across the sequence and fed into the router. In the decoder, the routers receive the average of the encoder's final hidden states instead of the decoder hidden states to prevent information leakage from later target tokens. We also include expert dropout Liu et al. (2022b) where each expert is dropped with a probability of 0.1 wherever it was found to be beneficial (a detailed ablation can be found in table 1). In GLUE, dataset sizes vary by three orders of magnitude, and we therefore found that load-balancing losses (as used e.g. in (Shazeer et al., 2017; Fedus et al., 2021; Lepikhin et al., 2020) to encourage uniform usage across experts) tended to hurt performance, so we did not include them.

**ResNet-DomainNet** In this setting, we focus on adapting an ImageNet pre-trained ResNet18 model (He et al., 2016) to datasets within DomainNet (Peng et al., 2019). DomainNet is a collection of object recognition datasets that cover six distinct domains and all share the same label space corresponding to 345 object categories. We treat the domain of each example as its tag. As in the T5-GLUE setting, we freeze the pretrained model and insert eight expert routing blocks after each of the eight residual layer groups in the model. Each block includes six experts corresponding to the number of domains. We use the same architecture for routers as in T5-GLUE and feed average-pooled hidden states into the router to compute the routing probability. Experts in this setting use batch normalization on their input instead of layer normalization in the output, following (Rebuffi et al., 2017). As in T5-GLUE, we omit load-balancing losses due to dramatically different sizes across domains in DomainNet.

Full details of hyperparameters and training timings for each setting are presented in appendix B.

**Results** To assess the overall effectiveness of routing strategies learned with SMEAR, we compare to learned routing using the gradient estimators, heuristic routing strategies, and modular baselines from section 2. A summary of our results is shown in fig. 2. First, we find that models using routing strategies learned through gradient estimation often underperform heuristic routing strategies – while the best-performing estimator (REINFORCE) in T5-GLUE outperforms tag routing, all estimators perform worse than tag routing in ResNet-DomainNet. On the other hand, we observed some cases

where gradient estimation-based routing outperforms hash or single-expert routing, which suggests that the learned routing strategies were nontrivial. Pertinently, in all experimental settings, SMEAR matches or outperforms every other routing strategy, including both routing learned by gradient estimators and all heuristic routing strategies. In particular, SMEAR achieves 2.7% improvement over tag routing in T5-GLUE and 0.6% improvement over tag routing in ResNet-DomainNet, suggesting effective specialization and sharing of experts. SMEAR additionally outperforms the single-expert parameter-matched baseline ($1\times$ parameters) by 1.4% in T5-GLUE and 1.2% in ResNet-DomainNet, further highlighting the importance of modularity.

As an upper bound on performance, we also compare SMEAR to expert ensembling ("Ensemble") which averages the outputs of all experts and incurs significantly higher computational cost. SMEAR matches the performance of ensemble routing in T5-GLUE and modestly underperforms it in ResNet-DomainNet, despite being significantly computationally cheaper. Compared to Adamix, which similarly averages experts but does not learn a routing strategy, SMEAR achieves 3.2% higher performance in T5-GLUE and 4% higher in ResNet-DomainNet. Since the Soft MoE method averages input tokens, it's inapplicable for the encoder-decoder model in T5-GLUE, where future tokens are not available for averaging in the decoder during inference. Hence, we include Soft MoE for ResNet-DomainNet where SMEAR outperforms it by 1.5%. Additionally, SMEAR exceeds the DSelect-$k$ method by 2.7% in ResNet-DomainNet. However, despite extensive hyperparameter tuning we encountered training instabilites with the DSelect-$k$ method in T5-GLUE and therefore omit those results. Moreover, while the performance improvement of SMEAR over Latent Skills is relatively small (0.6% in T5-GLUE and 0.1% in ResNet-DomainNet), a major advantage of SMEAR over Latent Skills is that it does not assume access to oracle tags (which are not always available in real-world settings) and instead learns an adaptive routing strategy. Finally, we highlight the consistency of improvements achieved by SMEAR across a diverse range of datasets and architectures, confirming its generality and robustness.

We additionally plot the inference speed (in terms of number of examples processed per second) of each method in fig. 2. The single-expert, Adamix, Hash, and Tag routing methods are the fastest since they do not use any routing networks. Despite the slight overhead of averaging the weights in SMEAR, we observe that its inference speed is almost identical to that of discrete adaptive routing (as learned via gradient estimation techniques). This confirms that the performance gains attained by SMEAR do not incur significant additional costs. Ensembling expert outputs is the slowest, with a $1.2\times$ slowdown in T5-GLUE and $1.3\times$ slowdown in ResNet-DomainNet compared to SMEAR.

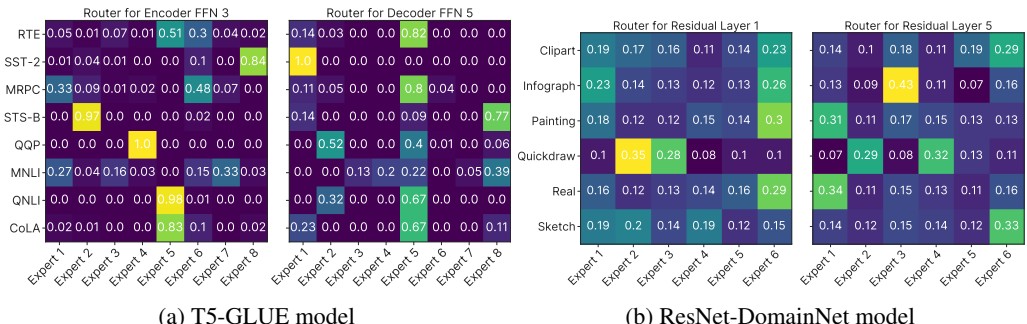

(a) T5-GLUE model          (b) ResNet-DomainNet model

Figure 3: Average routing distributions produced by SMEAR for two routers from the T5-GLUE model and two from the ResNet-DomainNet model. For a given router, we average all routing distributions across all examples from a given dataset.

**Scaling** Thus far, we have always set the number of experts equal to the number of tasks (in T5-GLUE) or domains (in DomainNet). However, with learned routing there is no reason to force this constraint, so we therefore tested the scalability of SMEAR by evaluating its performance with twice as many experts (16 for T5-GLUE and 12 for ResNet-DomainNet). We found a significant improvement (0.8%) when doubling the number of experts on ResNet-DomainNet, but no significant change on T5-GLUE ($81.3 \pm 1.1$ vs. $81.6 \pm 1.1$). This suggests there is no benefit to increasing capacity in the T5-GLUE setting. The complete results for doubling the number of experts are presented in appendix F (labeled "SMEAR $2\times$").

**Qualitative Analysis**  In this section, we provide qualitative analysis of the routing learned by SMEAR by visualizing the average router distribution across all examples in a given dataset for every router in each model. Figure 3 shows four select visualizations (two from a SMEAR-based model trained in T5-GLUE and two from ResNet-DomainNet). Across the two T5-GLUE router distributions shown in fig. 3, we observe significantly different behavior – one mainly follows a tag routing-style strategy whereas the other routes most datasets to the same expert. However, we note that the tag-style router utilizes shared experts for RTE, MRPC, and MNLI; notably, these tasks are somewhat similar in that they all involve determining similarity among pairs of sentences. In the single-expert-style router, STS-B (the only regression task) and SST-2 (which has a distinct output vocabulary) are given dedicated experts, and MNLI (a large and relatively challenging dataset) is routed through many different experts. More broadly, we highlight that there is generally a great deal of sparsity in the learned routing distributions, suggesting a significant amount of expert specialization. In ResNet-DomainNet, we can see that examples from the Quickdraw domain are routed to two specific experts in both cases. Additionally, we observe that the router distribution of the Painting and Real domains are highly correlated. Other domains such as Clipart and Sketch seem to evenly use experts. Interestingly, there is less expert specialization in the ResNet-DomainNet model, suggesting that there may be more similarities between the individual domains in DomainNet compared to the tasks in GLUE.

In general, other approaches for learning routing did not exhibit as intuitive or meaningful specialization and sharing. In ResNet-DomainNet, Top-$k$ demonstrates uniform routing in the initial layers but chooses a single expert in the last layer. REINFORCE, ST-Gumbel, and DSelect-$k$ tend to exhibit mostly degenerate single-expert routing. Interestingly, all these gradient estimators learn to assign a distinct expert for the Quickdraw dataset. However, this degree of specialization is insufficient for achieving superior performance scores. In T5-GLUE, these estimators display degenerate routing in some layers, while showing a tendency to share a few experts (approximately 3 out of 8) in other layers across tasks. Methods such as Latent Skills and Ensemble utilize most experts in the MoE layer (similar to SMEAR). Routing distribution visualizations for all methods and all layers can be found in appendix G.

## 5 RELATED WORK

**Models with Conditional Computation**  Various works have investigated ways learning discrete routing strategies. Deecke et al. (2020); Hazimeh et al. (2021); Dua et al. (2021) start training with most of the experts activated and gradually introduce sparsity. Kudugunta et al. (2021); Ponti et al. (2022); Ma et al. (2019); Gupta et al. (2022) group examples from the same task together and introduce task-specific parameters in the router. Other works avoid learned routing by hand-crafting heuristic routing strategies. Gururangan et al. (2021) built sparsely activated language models where different domains use separate experts and then weights the experts for new domains. Tang et al. (2022); Pfeiffer et al. (2022; 2020) assign experts based on task-related human knowledge. Our focus on settings where performant routing schemes can be hand-designed takes inspiration from this line of work. Because sparsely activated models disentangle computation and parameter count, significant effort has gone into leveraging conditional computation to create massive pre-trained models with a feasible computation cost (Fedus et al., 2022; Shazeer et al., 2017; Fedus et al., 2021; Du et al., 2022; Zoph et al., 2022; Yu et al., 2022). Many works explore different routing methods in this setting, with a major focus on balancing the load across experts (Lewis et al., 2021; Zhou et al., 2022; Kool et al., 2021; Roller et al., 2021). Another line of work aims to introduce ways to convert trained dense models into similar-sized sparse models with a lower computational footprint (Lee-Thorp & Ainslie, 2022; Zhang et al., 2022; Komatsuzaki et al., 2022). Previous studies have theoretically analyzed gradient estimators, focusing on the bias and variance of these gradients and suggesting enhancements through improved relaxation techniques Grathwohl et al. (2017), variance reduction via increased sampling Kool et al. (2019), and unbiased load balancing across experts Kool et al. (2021). The fundamental theoretical advantage of our method lies in its ability to enable exact gradient computation through standard backpropagation.

**Issues with Conditional Computation**  A great deal of past work has highlighted issues with models that use conditional computation. Clark et al. (2022) study the scaling laws of sparse language models and discovered a computational cutoff above which no additional benefits are observed. Relatedly, Du et al. (2022) observe worse results when further scaling up the number of experts.

Chi et al. (2022) highlight that using the model's activations as input to the router can cause the representations to "collapse". Dai et al. (2022) demonstrate that learned routing decisions can fluctuate significantly over training. Mittal et al. (2022) create a set of simple and compositional data distributions and show that systems with modular architecture can not find the most performant solution when trained end-to-end. Ye et al. (2022) experiment with various designs for multi-task learning with task-level routing and find that the performance never surpasses simple multi-task baselines. We show a possibility to avoid these issues with a fully differentiable routing strategy that does not increase computational costs.

**Weight Averaging Methods** Many prior works utilize parameter averaging for ensembling. Wortsman et al. (2022c); Ilharco et al. (2022) average the weights of a pre-trained and a fine-tuned model to improve performance on target tasks as well as robustness to distribution shift. Choshen et al. (2022b) similarly show that merging multiple models fine-tuned on different datasets can provide a better initialization than using the original pre-trained model for further fine-tuning on new unseen datasets. Yang et al. (2019); Zhang et al. (2021) compute convolution kernels by averaging weights of individual kernels. Since the convolution operation is linear, weight averaging and ensembling are mathematically equivalent. However, SMEAR performs averaging on non-linear and parameter efficient experts that, when trained alone, can match the performance of the fully fine-tuned model Houlsby et al. (2019). $\pi$-Tuning Wu et al. (2023) employs a set of existing task specific experts, retrieving the top $k$ experts for a downstream task and learns to interpolate among these experts for the downstream task. While $\pi$-Tuning enables transfer learning to a new downstream task by learning to interpolate, our focus is on developing a routing algorithm that learns how to share or specialize experts without using any metadata. Model averaging is also a common step in distributed optimization, where it is widely used in federated learning (McMahan et al., 2017) and has recently been used for distributed fine-tuning (Wortsman et al., 2022a), multi-domain training (Li et al., 2022), and multitask training (Don-Yehiya et al., 2022). There are also works that utilize different styles of merging instead of weight averaging of parameters, such as reweighting parameters in accordance with their approximate Fisher information (Matena & Raffel, 2021), aligning features by fitting a linear projection (Jin et al., 2022), and permuting columns to account for permutation symmetries (Ainsworth et al., 2022).

## 6 CONCLUSION

In this work, we sought to address shortcomings of models with discrete routing among experts that can lead them to underperform heuristic non-learned routing. We hypothesized that these issues stem from the gradient estimation techniques required to propagate gradients through discrete routing decisions and therefore focused on designing an expert routing architecture that allows exact calculation of gradients. Our approach, called SMEAR, works by computing a weighted average of expert parameters where the weighting is set according to the output of a learned router. We compared the performance of models using SMEAR to discrete routing models that were trained via various gradient estimation techniques. In experimental settings covering different modalities and model architectures, we found that SMEAR outperformed all models with discrete routing as well as performant heursitic routing strategies. Notably, this performance boost comes with no increase in computational costs. SMEAR also matched or outperformed existing state-of-the-art methods for learning modular models through expert averaging while removing the requirement for oracle task labels. Through qualitative analysis, we further confirmed that the experts learned in a model using SMEAR specialize to different types of inputs and that the router learns a nontrivial strategy that exploits commonalities across different examples. In future work, we are interested in exploring different expert architectures (Liu et al., 2022a; Hu et al., 2021) and improved merging methods (Matena & Raffel, 2021; Ainsworth et al., 2022; Jin et al., 2022). Given access to a larger amount of compute, we would also be excited to try out SMEAR in the large-scale settings where discrete routing has been used (Fedus et al., 2021; Zoph et al., 2022; Du et al., 2022) to see whether it helps fix the poor scaling properties of models with discrete routing (Clark et al., 2022).

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

## A    COMPUTE RESOURCES USED

Here we provide details on the compute resources used in our experiments. All models were trained on 48GB A6000s, except for the Ensemble method, which was trained on 80GB A100s. The training time for each T5-GLUE experiment was approximately 108 hours while each ResNet-DomainNet experiment required approximately 11 hours of training.

## B    EXPERIMENT DETAILS

In this section, we provide details on the experimental setup and hyperparameter choices for the T5-GLUE and ResNet-DomainNet experiments described in the main text. We implemented Adamix Wang et al. (2022b) and ran with the hyperparameters listed in the below subsections. In Latent Skills Ponti et al. (2022), we use Adapters consistent with all other experiments and chose learning rate ratio of 10 for skill matrix, which we found to be best after sweeping for $\{1, 10, 100\}$ in both the settings. For exact implementation details of above methods, we refer the reader to each of the respective works.

It is also important to mention that SMEAR's memory footprint is comparable to that of other methods. Since each expert's size is moderate, we compute expert outputs by preparing an expert for each example and applying them to the examples in parallel. Thus, none of the methods need an extra weight-sized tensor. Training on T5-GLUE requires around 30GB of memory for SMEAR and other methods, with no significant differences observed in ResNet-DomainNet.

### B.1    T5-GLUE

GLUE consists of nine datasets (SST-2 (Socher et al., 2013), CoLA (Warstadt et al., 2019)), MNLI (Williams et al., 2017), RTE (Bentivogli et al., 2009), QQP (Shankar et al., 2017), MRPC (Dolan & Brockett, 2005), STS-B (Cer et al., 2017), QNLI (Rajpurkar et al., 2016), and WNLI (Levesque et al., 2012)) that cover a wide range of natural language processing tasks. Following convention, we exclude WNLI and use the remaining eight datasets. We use the prompted form of these datasets available in PromptSource (Bach et al., 2022), which maps each example into a natural language request-and-response form. During training, we randomly select a prompt templates for each example; during evaluation, we evaluate each example using all of its dataset's templates. In the T5-GLUE experiments in this paper, we concatenated all 8 datasets of GLUE and perform multitask training. T5 models were trained for $600k$ steps using a learning rate of $3e^{-4}$, with $2k$ warmup steps, and batch size of 128. The AdamW optimizer was used with its default settings. We ran the ST-Gumbel estimator with a $\tau$ value of 10 and an anneal rate of $1e^{-6}$ by sweeping $\tau$ in the range of $\{1, 10\}$ and the anneal rate in the range of $\{1e^{-4}, 1e^{-5}, 1e^{-6}\}$. For the REINFORCE estimator, we used the same values as in (Clark et al., 2022), $\alpha = 1e^{-2}$, $\beta = 5e^{-4}$, and $\gamma = 1e^{-2}$. The adapters here use swish non-linearity in between.

### B.2    RESNET-DOMAINNET

In the ResNet-DomainNet experiments, all the domains from DomainNet were concatenated to perform multitask training similar to T5-GLUE. ResNet models were trained for $100k$ steps with batch size of 128 and a learning rate of $1e^{-3}$, with no warm up, using Adam optimizer. We used $\tau$ value of 10 and anneal rate of $1e^{-4}$ for the ST-Gumbel estimator by sweeping $\tau$ in the range of $\{1, 10\}$ and the anneal rate in the range of $\{1e^{-4}, 1e^{-5}, 1e^{-6}\}$. The values of $\alpha$, $\beta$, and $\gamma$ for the REINFORCE estimators are same as in T5-GLUE experiments. The hyperparameter that weighs entropy regularization in Dselect-$k$ is chosen as $0.1$ which worked best among $\{0.01, 0.1, 1\}$. The adapters also used a swish non-linearity in between.

## C    EXPERT DROPOUT

Table 1 illustrates the impact of expert dropout on methods that do not make use of metadata but learn adaptive routing, namely methods that are trained through gradient estimation and SMEAR. We conduct this ablation on a single seed to limit the amount of computation. The results show that

SMEAR benefits from an improvement of 2.9% on T5-GLUE, and Top-$k$ achieves a 1.2% and 0.2% improvement on T5-GLUE and ResNet-DomainNet respectively. As a result, we include expert dropout for these two methods when discussed in the main text. However, expert dropout has a negative impact on the performance of ST-Gumbel and REINFORCE methods, and thus, we exclude it for these two methods in the main text.

| Routing | T5-GLUE | ResNet-DomainNet |
|---|---|---|
| Top-$k$ | 77.2 | 59.8 |
| w/ Expert dropout 0.1 | 78.4 (+1.2) | 60.0 (+0.2) |
| ST-Gumbel | 78.3 | 58.3 |
| w/ Expert dropout 0.1 | 77.2 (-1.1) | 57.9 (-0.4) |
| REINFORCE | 79.8 | 59.8 |
| w/ Expert dropout 0.1 | 78.2 (-1.6) | 59.8 (+0.0) |
| SMEAR | 80.2 | 62.0 |
| w/ Expert dropout 0.1 | 83.1 (+2.9) | 62.0 (+0.0) |

Table 1: Performance comparision of different adaptive routing methods w and w/o dropout on a single seed. The results indicate that SMEAR and Top-$k$ method benefit from the expert dropout, while ST-Gumbel and REINFORCE are negatively affected.

## D    LICENSE

T5 is licensed under Apache 2.0. The ResNet model we used is licensed under BSD 3-Clause License. QNLI uses CC BY-SA 4.0 license. MultiNLI uses data sources of multiple different licenses(Williams et al., 2017). CoLA, SST-2, RTE, MRPC, STS-B, QQP, and DomainNet allow non-commercial research use cases. Our code for T5-GLUE is based on HyperformerMahabadi et al. (2021), which is shared under Apache 2.0. The code for ResNet-DomainNet is developed by us.

## E    ETHICS STATEMENT

We are not aware of any negative ethical implications of our work. Our work does not involve human subjects and is primarily focused on diagnosing issues with an efficient class of neural networks. While conditional computation has been used to design extremely large neural networks (Shazeer et al., 2017; Fedus et al., 2021; Du et al., 2022) that have high computational costs (and, correspondingly, energy usage), our work primarily focuses on smaller-scale models.

## F    FULL RESULTS ON T5-GLUE AND RESNET-DOMAINNET

We show the full results of T5-GLUE in table 2 and ResNet-DomainNet in table 3.

## G    ROUTING DISTRIBUTION IN ALL ROUTING BLOCKS

Here, we present the routing distribution across all routing blocks in both T5-GLUE and ResNet-DomainNet as learned by SMEAR. This is depicted in fig. 4, fig. 5, fig. 6, and fig. 7.

For other learned routing methods, refer to the corresponding figures for detailed routing distributions:

- **Latent Skills**: fig. 8, fig. 9, fig. 10, and fig. 11.
- **Top-$k$**: fig. 12, fig. 13, fig. 14, and fig. 15.
- **ST-Gumbel**: fig. 16, fig. 17, fig. 18, and fig. 19.
- **REINFORCE**: fig. 20, fig. 21, fig. 22, and fig. 23.
- **Dselect-$k$**: fig. 24
- **Ensemble**: fig. 25, fig. 26, fig. 27, and fig. 28.

| Routing | RTE acc | SST-2 acc | MRPC f1 | MRPC acc | STS-B pearson | STS-B spearman | QQP f1 | QQP acc | MNLI acc | QNLI acc | CoLA mcc | Average |
|---|---|---|---|---|---|---|---|---|---|---|---|---|
| SMEAR | $69.9_{2.6}$ | $90.9_{0.8}$ | $90.5_{1.5}$ | $86.9_{2.2}$ | $87.0_{0.7}$ | $86.6_{0.8}$ | $86.9_{0.3}$ | $90.1_{0.2}$ | $84.9_{0.5}$ | $90.2_{0.6}$ | $33.8_{6.4}$ | $81.6_{1.0}$ |
| $1\times$ parameters | $72.3_{2.1}$ | $92.1_{0.5}$ | $89.9_{0.5}$ | $86.0_{0.8}$ | $85.5_{0.8}$ | $85.3_{0.9}$ | $87.0_{0.3}$ | $90.2_{0.2}$ | $84.1_{0.5}$ | $89.9_{0.7}$ | $20.1_{8.1}$ | $80.2_{0.8}$ |
| $1\times$ compute | $67.3_{3.3}$ | $91.9_{0.2}$ | $89.2_{2.7}$ | $85.5_{3.4}$ | $87.4_{0.8}$ | $87.3_{0.7}$ | $85.6_{0.3}$ | $89.2_{0.2}$ | $84.5_{0.7}$ | $89.9_{0.5}$ | $5.1_{3.7}$ | $78.4_{1.1}$ |
| Adamix | $70.2_{3.2}$ | $92.4_{0.6}$ | $87.4_{1.4}$ | $83.3_{1.2}$ | $86.7_{0.6}$ | $86.6_{0.7}$ | $85.7_{0.2}$ | $89.4_{0.1}$ | $85.4_{0.3}$ | $90.5_{0.4}$ | $5.1_{1.1}$ | $78.4_{0.4}$ |
| Hash | $58.8_{2.7}$ | $85.6_{1.3}$ | $77.7_{2.6}$ | $68.5_{3.2}$ | $65.4_{2.4}$ | $65.2_{1.8}$ | $76.8_{0.2}$ | $82.8_{0.3}$ | $72.0_{0.9}$ | $80.0_{0.5}$ | $2.9_{2.5}$ | $66.9_{0.9}$ |
| Tag | $71.7_{2.9}$ | $90.3_{0.5}$ | $85.4_{0.7}$ | $79.5_{0.8}$ | $82.2_{1.1}$ | $81.5_{1.2}$ | $86.2_{0.4}$ | $89.5_{0.3}$ | $84.4_{0.8}$ | $87.9_{0.9}$ | $25.1_{8.3}$ | $78.5_{1.2}$ |
| Latent Skills | $70.4_{4.6}$ | $90.8_{0.8}$ | $88.6_{0.7}$ | $85.8_{2.1}$ | $86.6_{1.5}$ | $86.3_{1.4}$ | $86.4_{0.4}$ | $89.8_{0.3}$ | $84.9_{0.9}$ | $89.3_{1.4}$ | $30.8_{5.5}$ | $81.0_{1.6}$ |
| Top-$k$ | $68.2_{2.3}$ | $92.5_{0.4}$ | $88.8_{1.0}$ | $84.7_{1.7}$ | $87.7_{1.6}$ | $87.4_{1.7}$ | $85.3_{0.4}$ | $89.0_{0.5}$ | $84.9_{0.9}$ | $90.1_{0.9}$ | $2.0_{2.0}$ | $78.2_{0.9}$ |
| ST-Gumbel | $67.6_{2.3}$ | $92.1_{0.7}$ | $88.1_{0.1}$ | $84.8_{1.7}$ | $86.9_{1.0}$ | $86.8_{0.8}$ | $85.7_{0.1}$ | $89.2_{0.2}$ | $84.5_{0.3}$ | $89.1_{0.5}$ | $1.3_{1.7}$ | $77.9_{0.4}$ |
| REINFORCE | $70.9_{3.3}$ | $92.6_{0.5}$ | $89.8_{1.6}$ | $86.0_{2.1}$ | $87.4_{0.6}$ | $87.2_{0.5}$ | $86.1_{0.3}$ | $89.5_{0.2}$ | $85.8_{0.4}$ | $90.8_{0.7}$ | $14.1_{6.9}$ | $80.0_{0.8}$ |
| Ensemble | $72.9_{1.6}$ | $91.5_{0.4}$ | $90.9_{1.4}$ | $87.7_{1.4}$ | $85.7_{1.5}$ | $85.1_{1.6}$ | $86.8_{0.3}$ | $90.1_{0.2}$ | $84.7_{0.5}$ | $89.8_{0.6}$ | $33.7_{6.1}$ | $81.7_{1.0}$ |
| SMEAR $2\times$ | $70.9_{3.1}$ | $90.9_{0.7}$ | $89.5_{1.1}$ | $85.8_{1.0}$ | $86.9_{0.9}$ | $86.5_{1.0}$ | $86.8_{0.4}$ | $90.1_{0.3}$ | $84.4_{0.5}$ | $89.6_{0.8}$ | $33.1_{6.5}$ | $81.3_{1.1}$ |

Table 2: Full T5-GLUE results.

| Routing | Clipart | Infograph | Painting | Quickdraw | Real | Sketch | Final Accuracy |
|---|---|---|---|---|---|---|---|
| SMEAR | $64.2_{0.1}$ | $31.2_{0.3}$ | $57.8_{0.3}$ | $62.3_{0.1}$ | $74.3_{0.1}$ | $56.0_{0.2}$ | $62.0_{0.1}$ |
| $1\times$ parameters | $63.3_{0.3}$ | $29.8_{0.3}$ | $56.4_{0.3}$ | $61.5_{0.1}$ | $72.9_{0.1}$ | $54.9_{0.4}$ | $60.8_{0.1}$ |
| $1\times$ compute | $60.2_{0.3}$ | $27.9_{0.3}$ | $54.8_{0.1}$ | $59.0_{0.2}$ | $72.3_{0.1}$ | $52.6_{0.2}$ | $59.0_{0.1}$ |
| Adamix | $58.9_{0.2}$ | $27.0_{0.2}$ | $54.1_{0.2}$ | $57.2_{0.3}$ | $72.1_{0.1}$ | $51.2_{0.2}$ | $58.0_{0.2}$ |
| Hash | $53.5_{0.3}$ | $23.4_{0.3}$ | $49.8_{0.4}$ | $48.6_{0.3}$ | $68.5_{0.1}$ | $45.7_{0.2}$ | $52.4_{0.1}$ |
| Tag | $62.8_{0.4}$ | $30.2_{0.3}$ | $58.0_{0.2}$ | $61.7_{0.2}$ | $74.1_{0.1}$ | $55.1_{0.3}$ | $61.4_{0.1}$ |
| Latent Skills | $64.5_{0.4}$ | $31.2_{0.4}$ | $58.9_{0.1}$ | $61.6_{0.3}$ | $74.2_{0.1}$ | $56.3_{0.2}$ | $61.9_{0.2}$ |
| Top-$k$ | $61.6_{0.2}$ | $29.6_{0.2}$ | $55.8_{0.4}$ | $60.2_{0.3}$ | $73.0_{0.2}$ | $53.5_{0.1}$ | $60.0_{0.1}$ |
| ST-Gumbel | $59.9_{0.3}$ | $27.6_{0.4}$ | $54.5_{0.3}$ | $58.1_{0.5}$ | $72.1_{0.2}$ | $51.9_{0.4}$ | $58.5_{0.2}$ |
| REINFORCE | $61.3_{0.3}$ | $29.1_{0.2}$ | $55.9_{0.2}$ | $60.4_{0.3}$ | $72.8_{0.1}$ | $53.6_{0.2}$ | $60.0_{0.1}$ |
| DSelect-$k$ | $60.6_{0.2}$ | $28.4_{0.3}$ | $55.1_{0.3}$ | $59.5_{0.3}$ | $72.5_{0.2}$ | $52.5_{0.4}$ | $59.3_{0.2}$ |
| Soft MoE | $62.7_{0.2}$ | $29.3_{0.3}$ | $56.5_{0.2}$ | $60.3_{0.2}$ | $73.3_{0.1}$ | $54.9_{0.1}$ | $60.5_{0.1}$ |
| Ensemble | $65.7_{0.1}$ | $32.3_{0.0}$ | $58.5_{0.3}$ | $63.7_{0.2}$ | $74.6_{0.1}$ | $57.6_{0.2}$ | $62.9_{0.1}$ |
| SMEAR $2\times$ | $65.3_{0.1}$ | $31.8_{0.1}$ | $58.4_{0.5}$ | $63.3_{0.3}$ | $74.9_{0.2}$ | $57.3_{0.1}$ | $62.8_{0.1}$ |

Table 3: Full ResNet-DomainNet results.

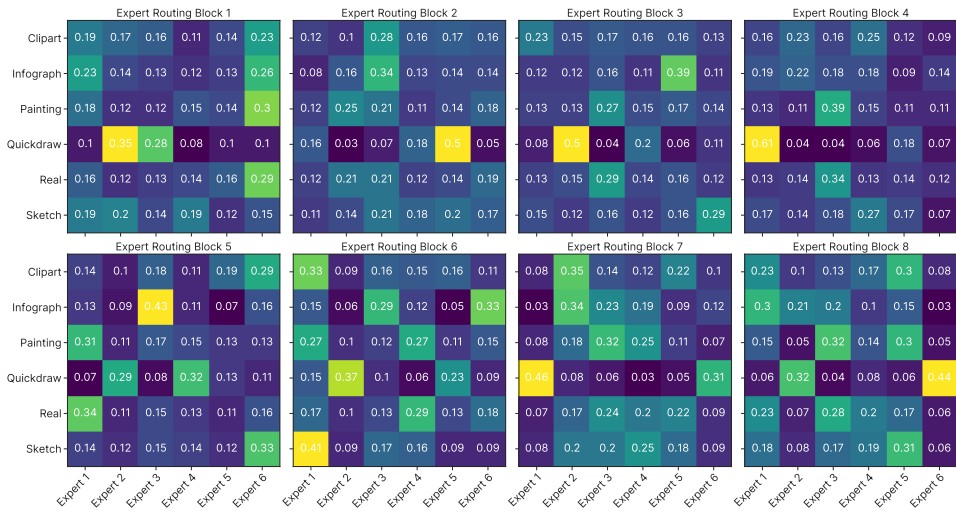

Figure 4: Routing distribution learnt by SMEAR in the routing blocks of ResNet-DomainNet

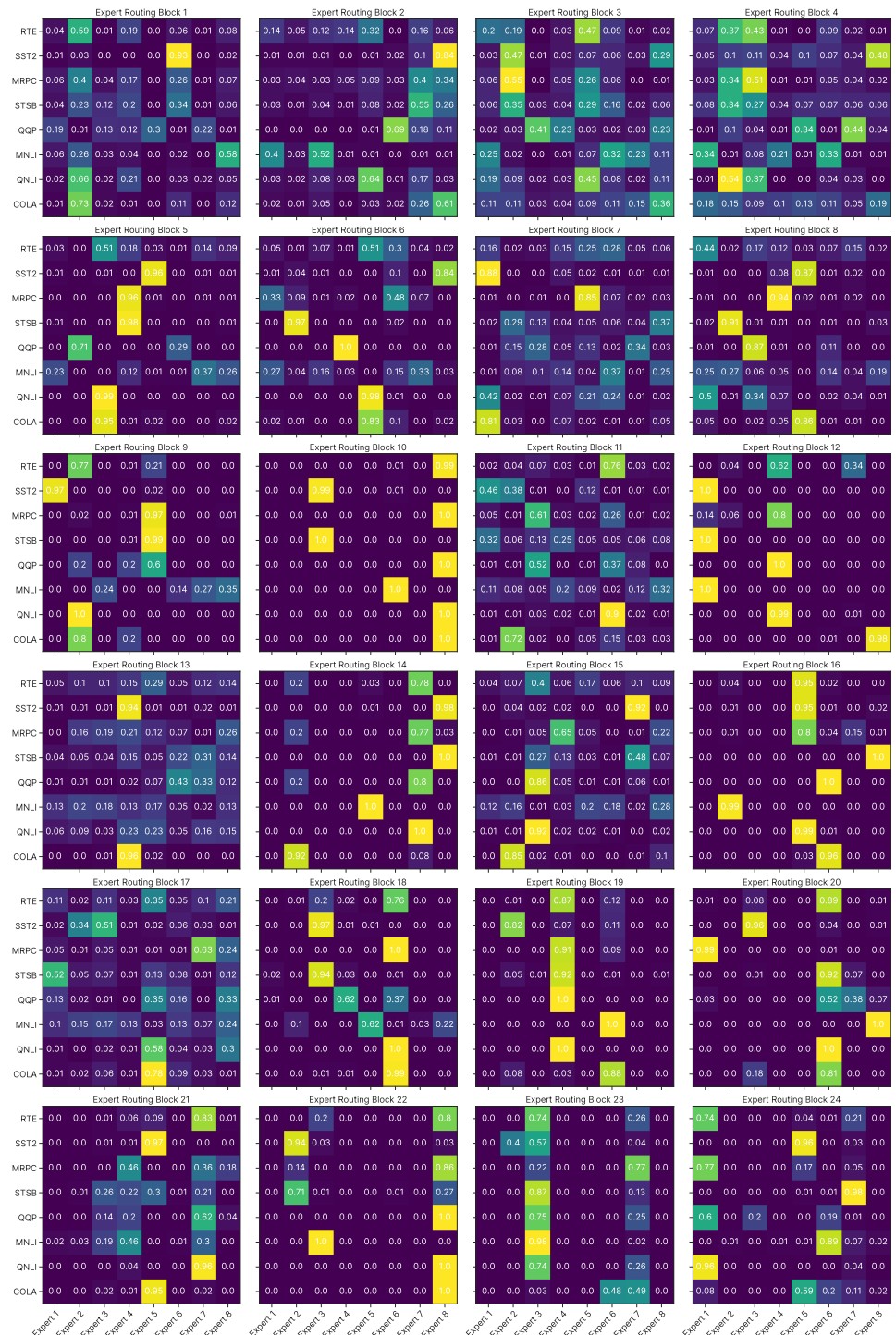

Figure 5: Routing distribution learnt by SMEAR in the encoder routing blocks (1-24) of T5-GLUE

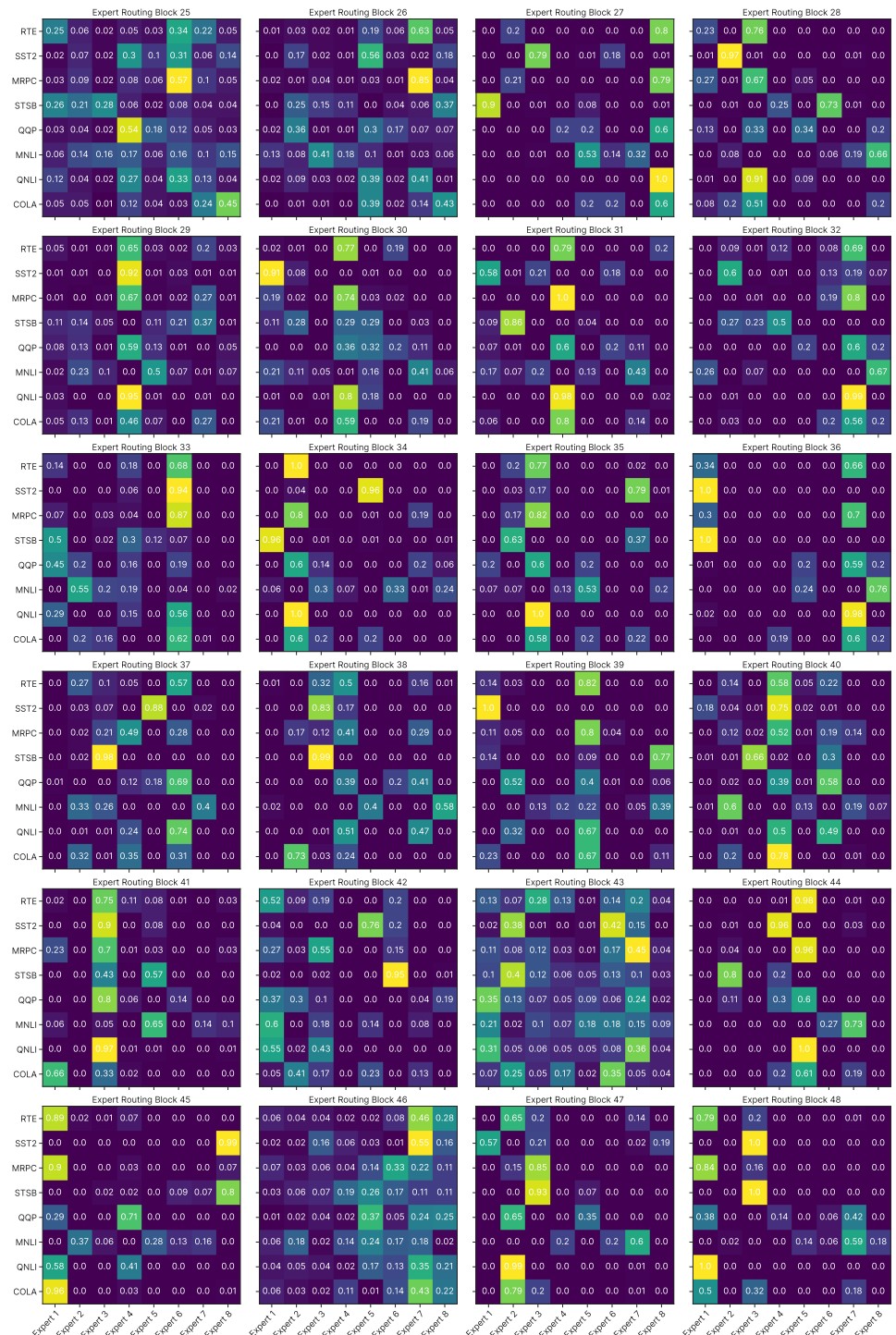

Figure 6: Routing distribution learnt by SMEAR in the decoder routing blocks (25-48) of T5-GLUE

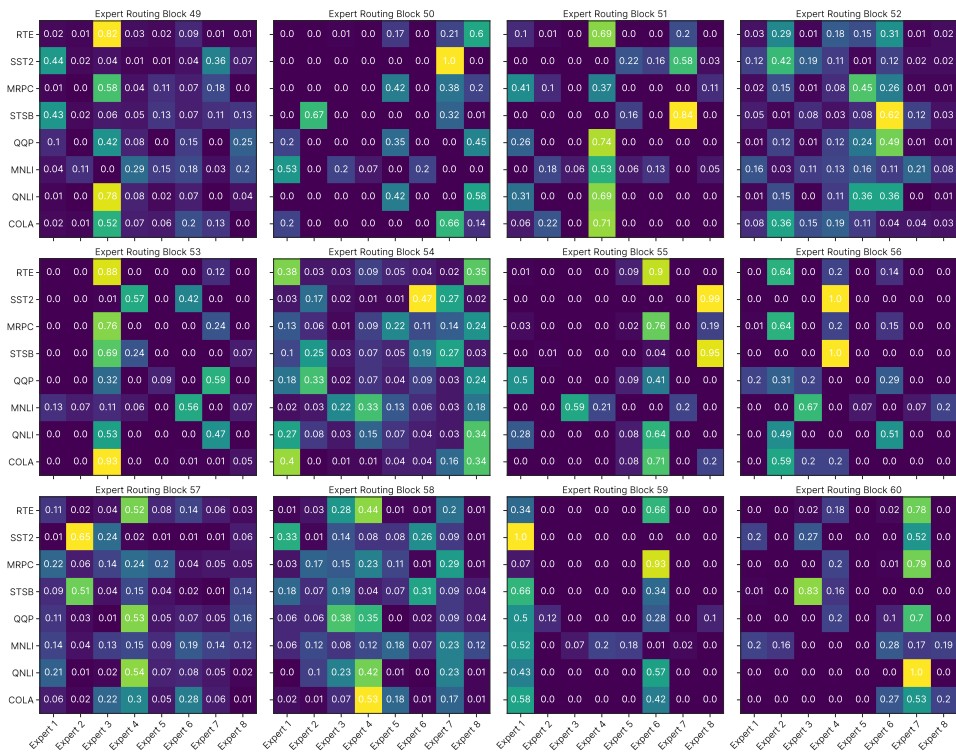

Figure 7: Routing distribution learnt by SMEAR in the decoder routing blocks (49-60) of T5-GLUE

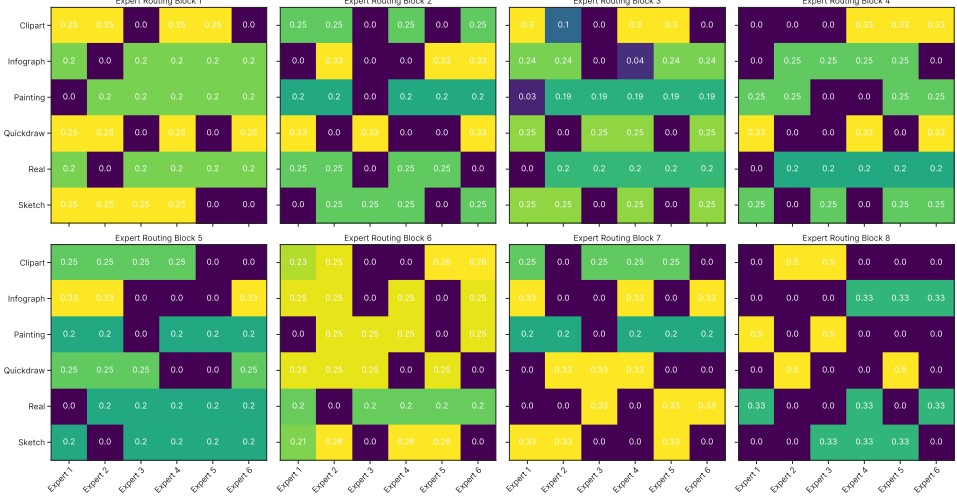

Figure 8: Routing distribution learnt by Latent Skills routing in the routing blocks of ResNet-DomainNet

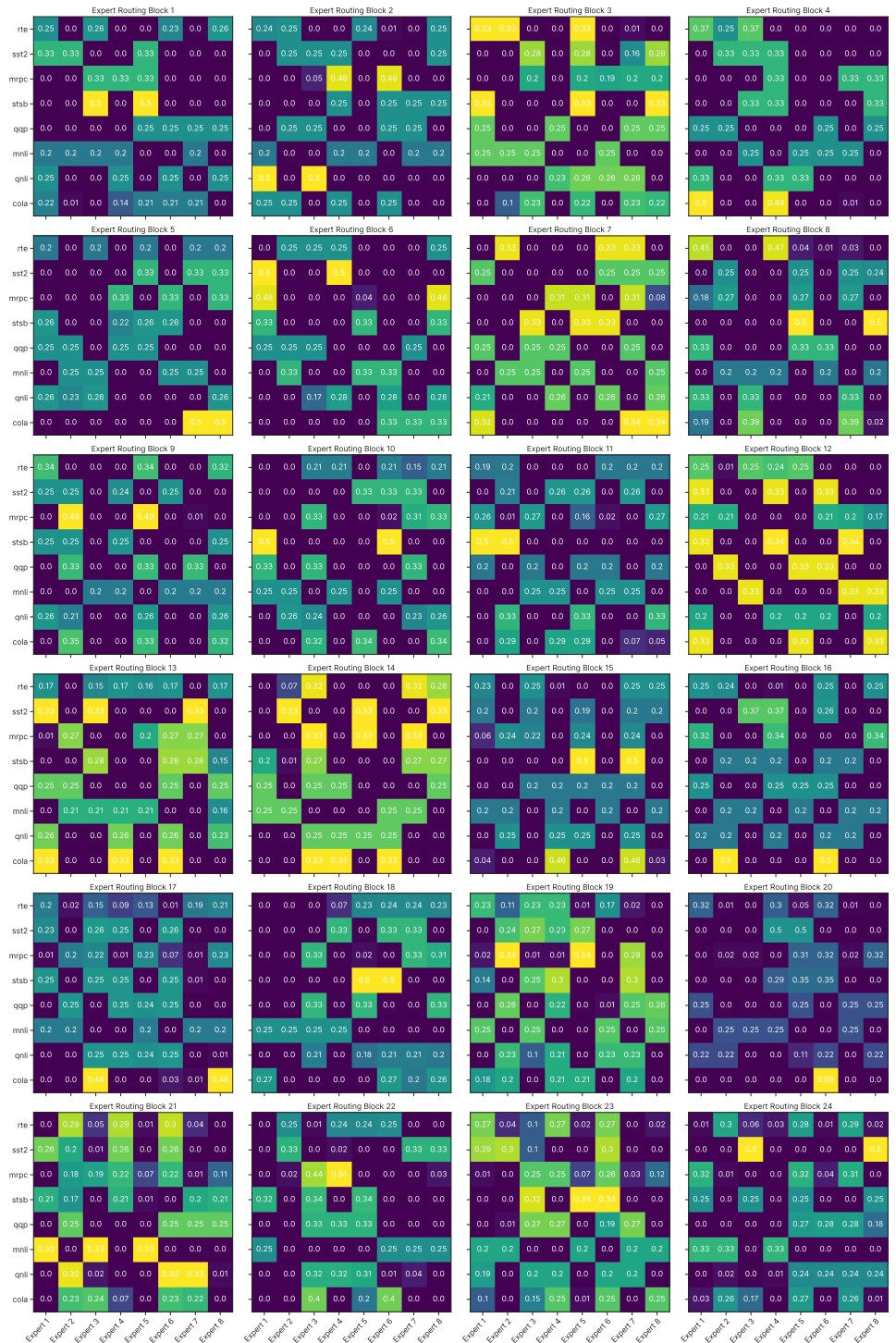

Figure 9: Routing distribution learnt by Latent Skills in the encoder routing blocks (1-24) of T5-GLUE

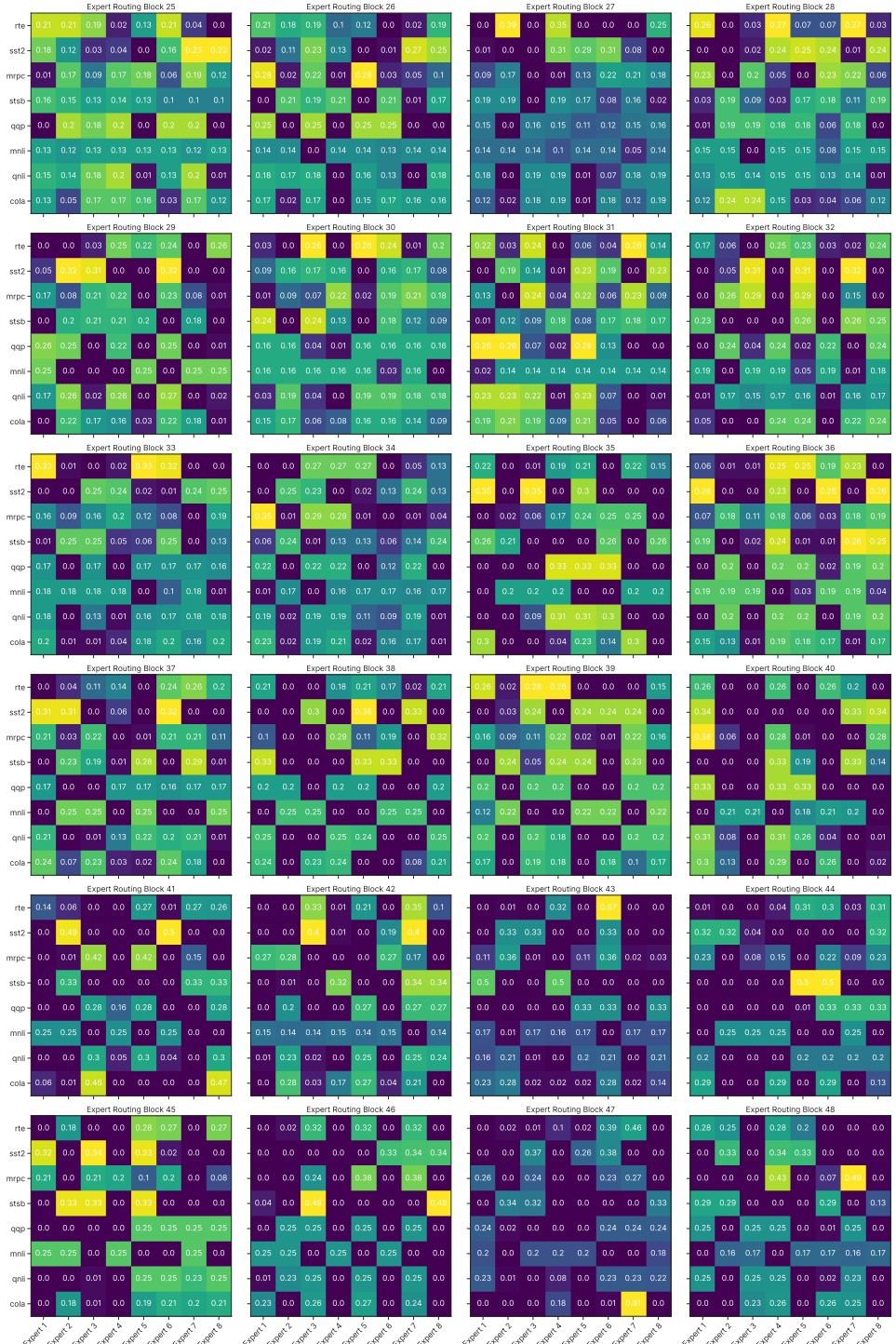

Figure 10: Routing distribution learnt by Latent Skills in the decoder routing blocks (25-48) of T5-GLUE

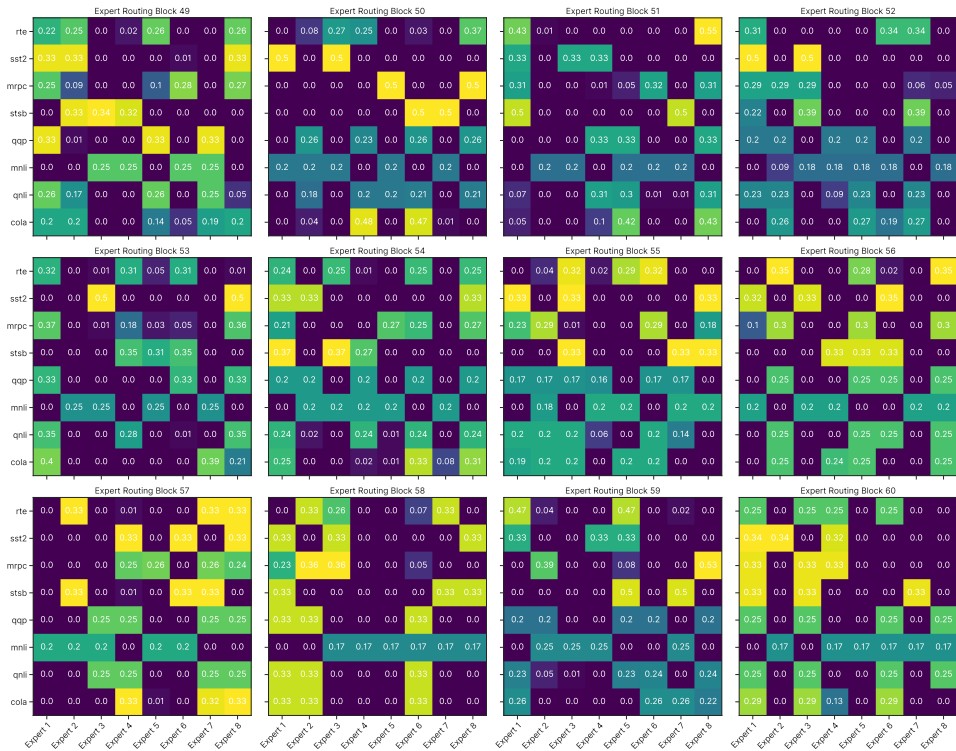

Figure 11: Routing distribution learnt by Latent Skills in the decoder routing blocks (49-60) of T5-GLUE

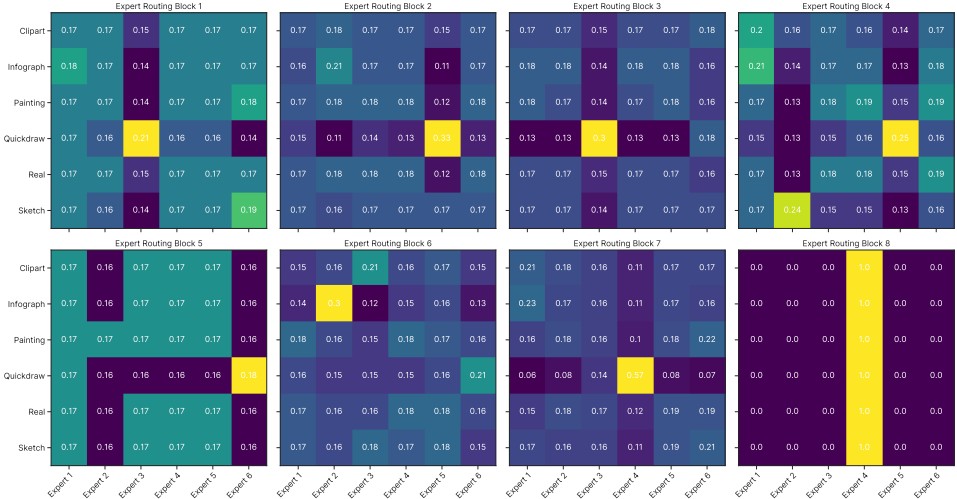

Figure 12: Routing distribution learnt by Top-$k$ routing in the routing blocks of ResNet-DomainNet

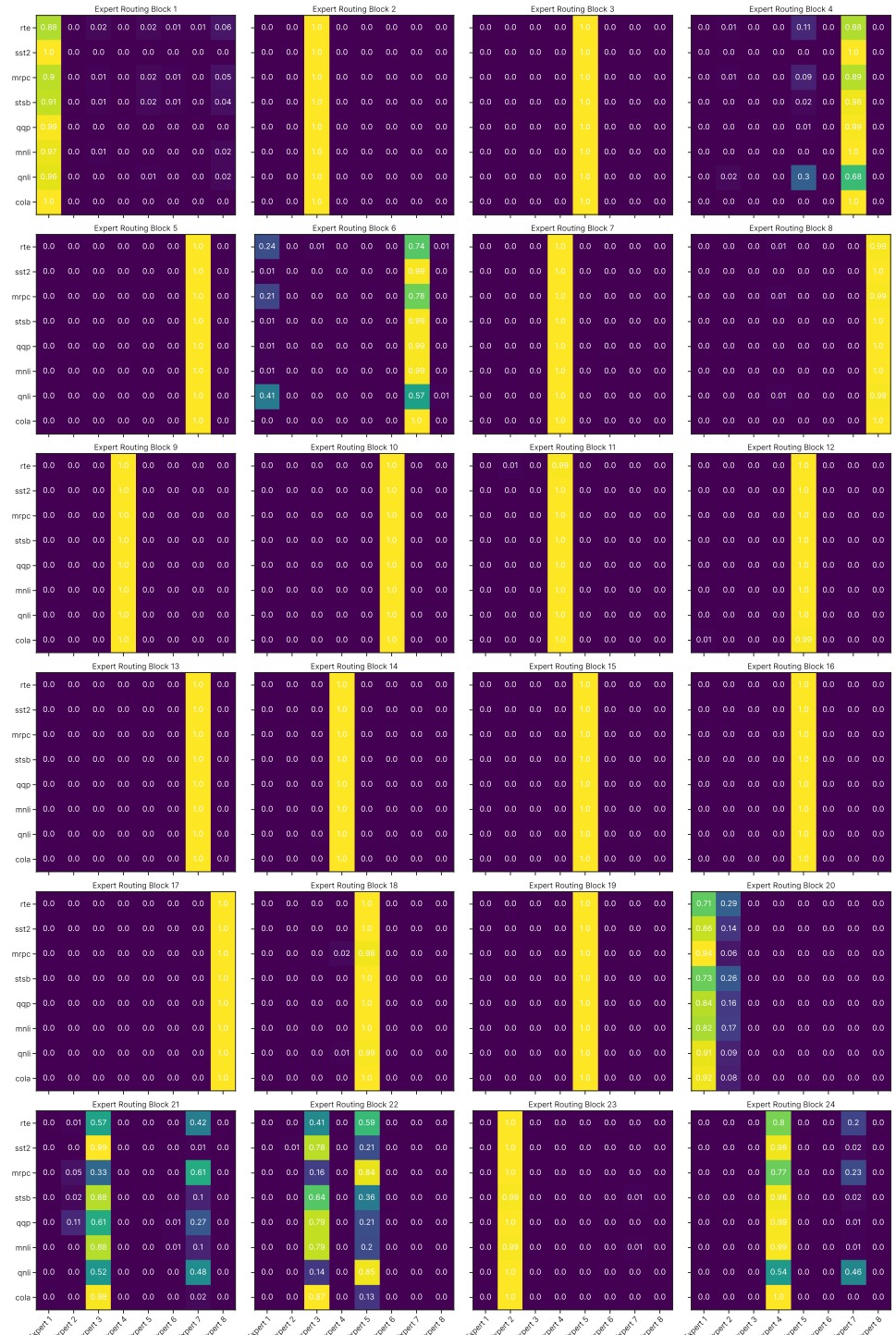

Figure 13: Routing distribution learnt by Top-$k$ in the encoder routing blocks (1-24) of T5-GLUE

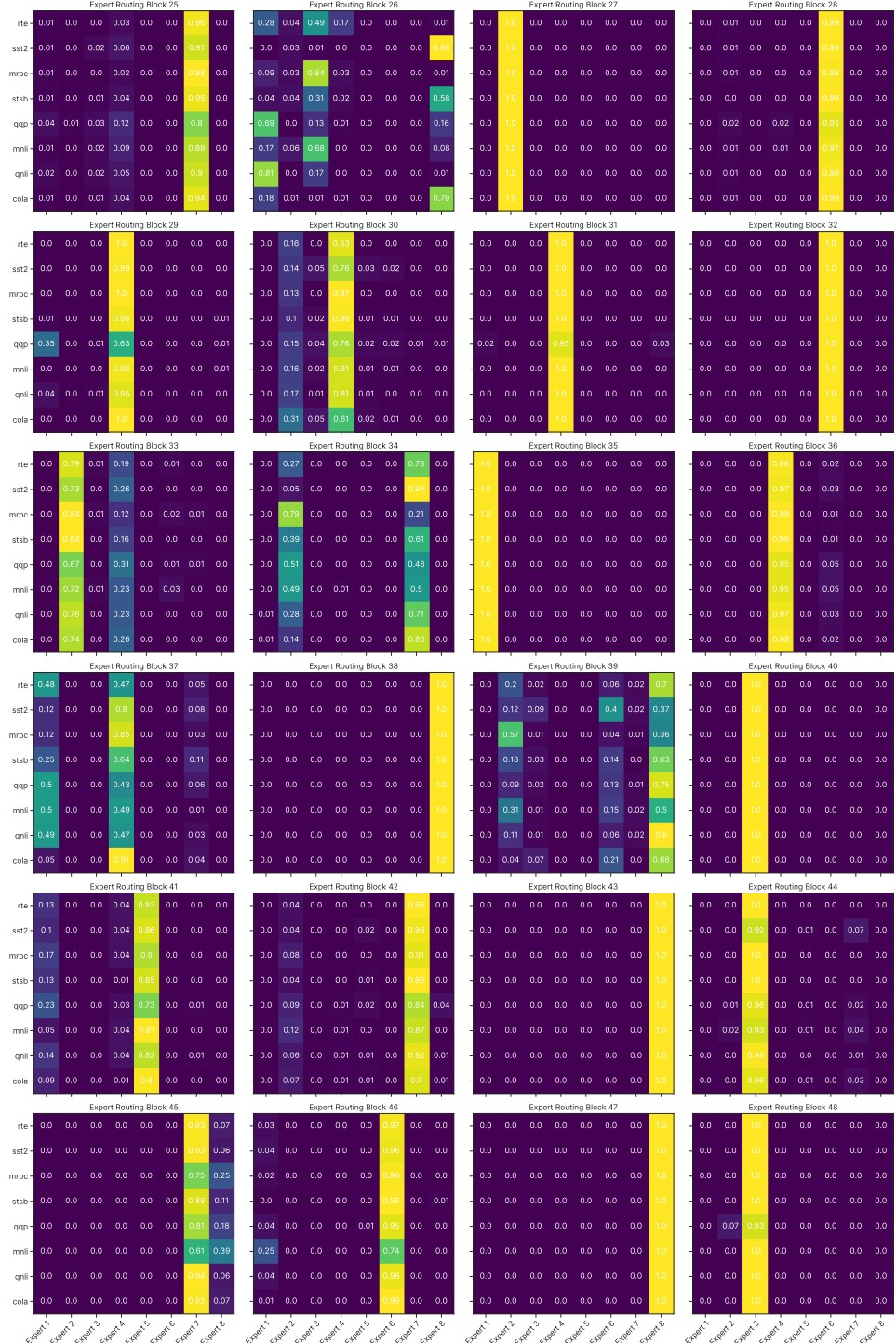

Figure 14: Routing distribution learnt by Top-$k$ in the decoder routing blocks (25-48) of T5-GLUE

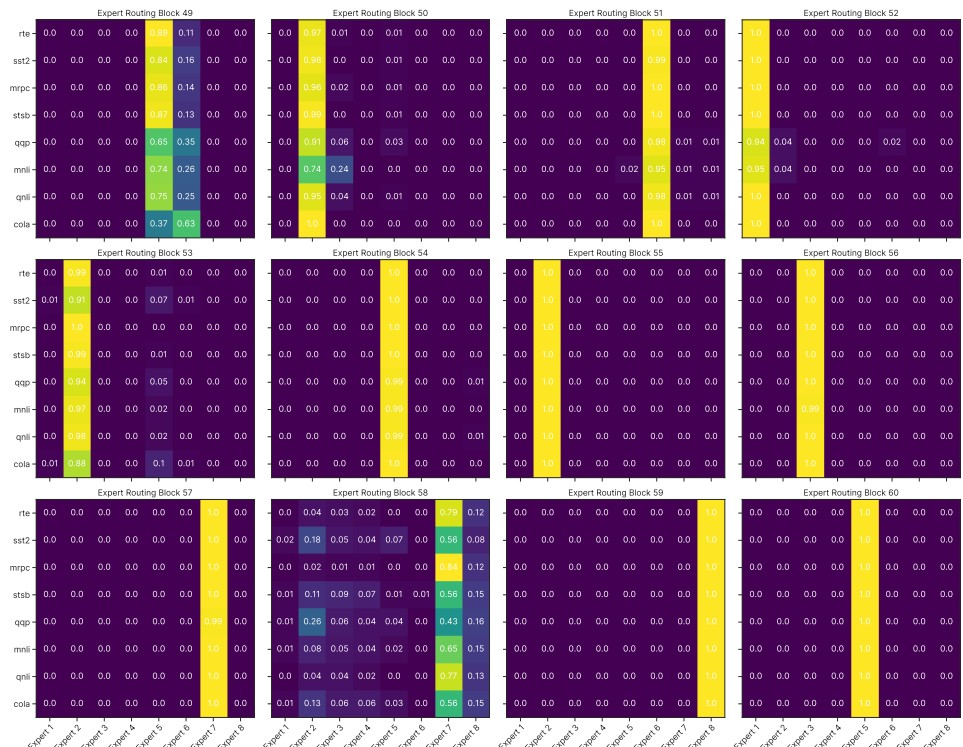

Figure 15: Routing distribution learnt by Top-$k$ in the decoder routing blocks (49-60) of T5-GLUE

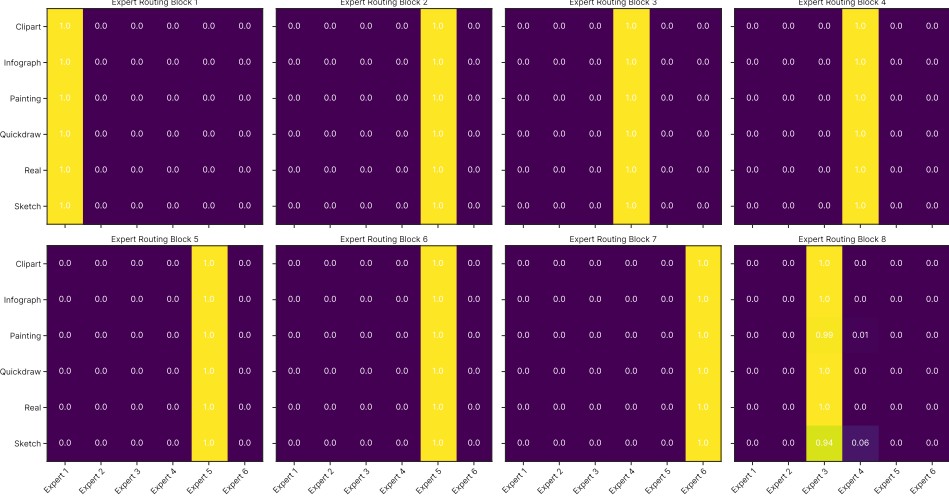

Figure 16: Routing distribution learnt by ST-Gumbel routing in the routing blocks of ResNet-DomainNet

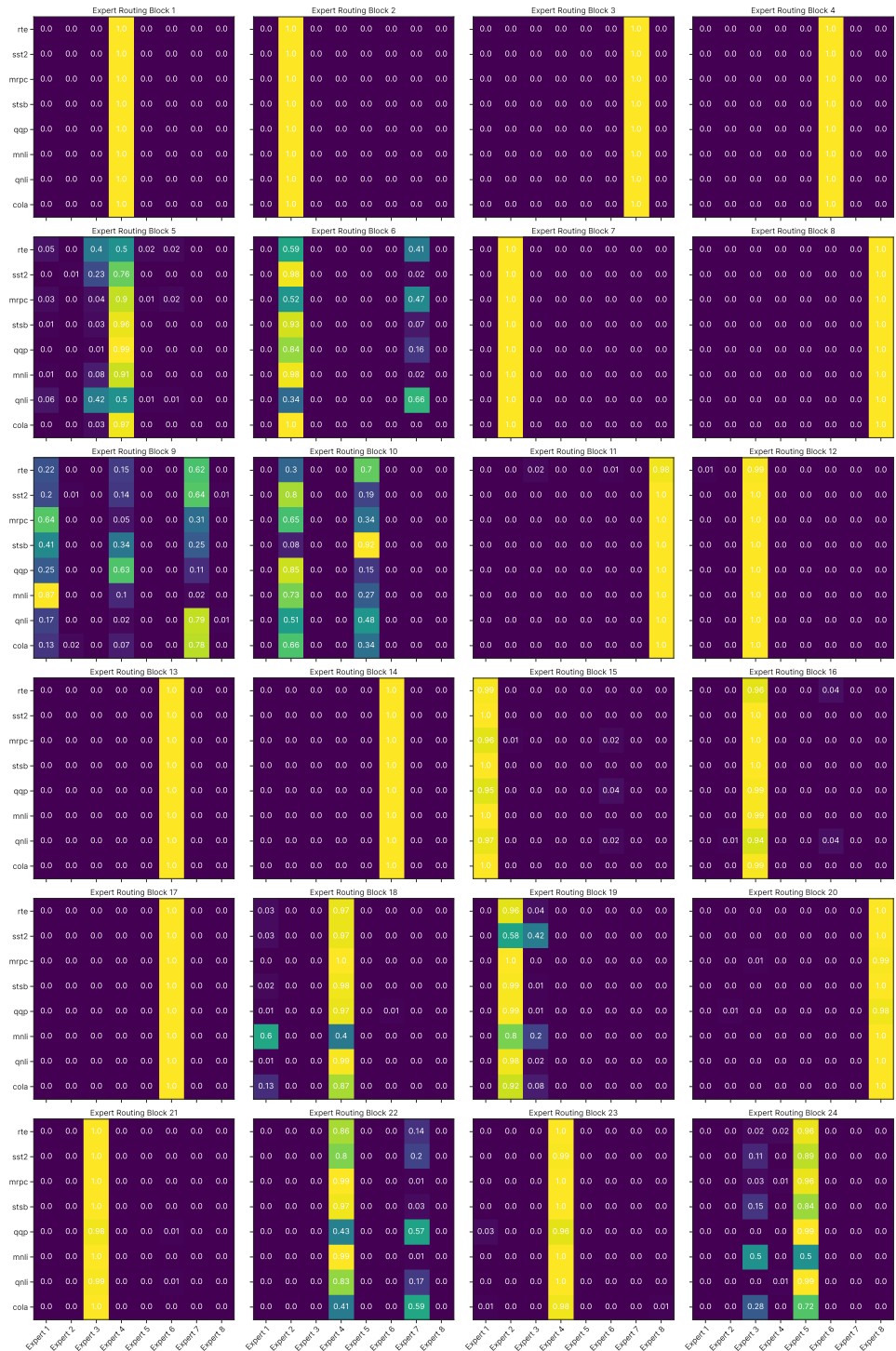

Figure 17: Routing distribution learnt by ST-Gumbel in the encoder routing blocks (1-24) of T5-GLUE

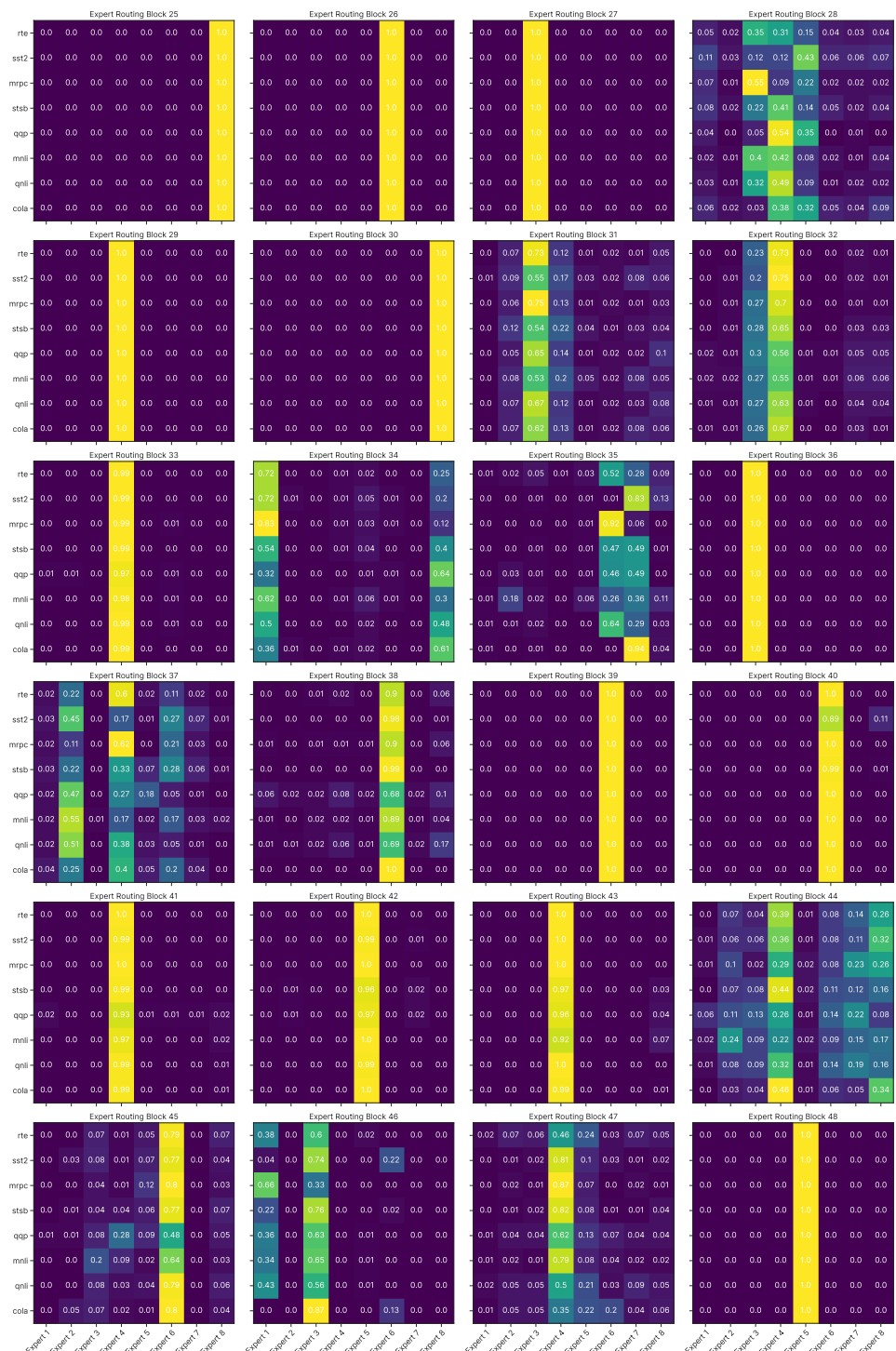

Figure 18: Routing distribution learnt by ST-Gumbel in the decoder routing blocks (25-48) of T5-GLUE

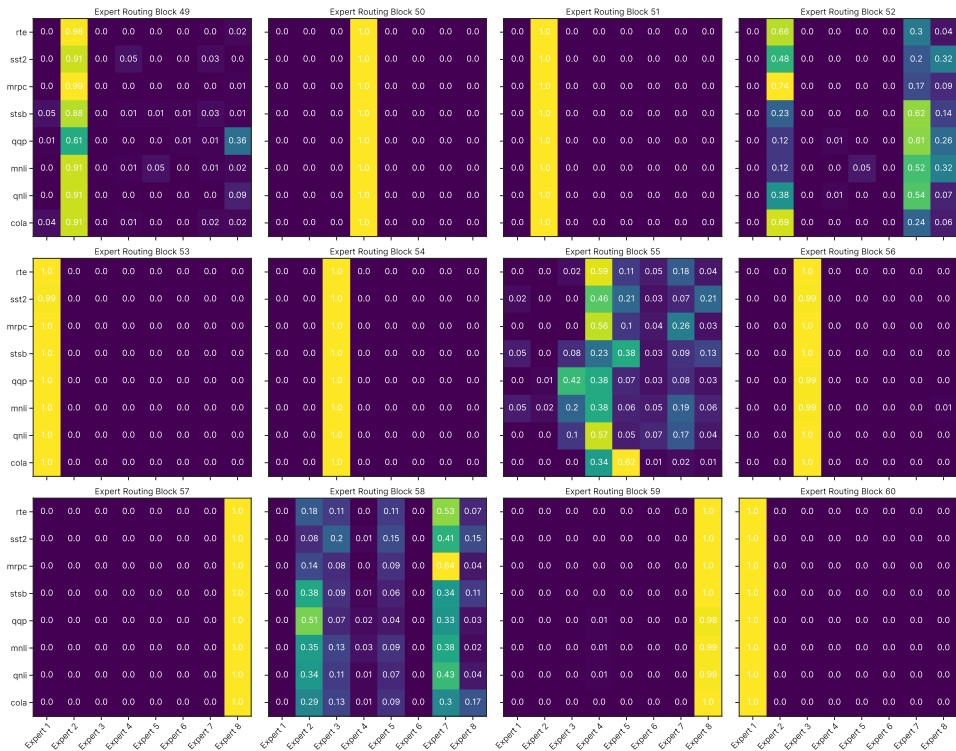

Figure 19: Routing distribution learnt by ST-Gumbel in the decoder routing blocks (49-60) of T5-GLUE

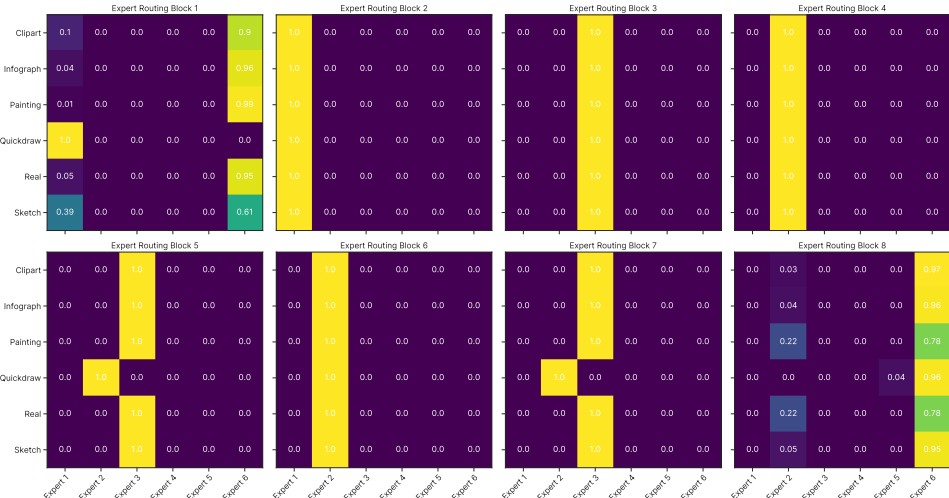

Figure 20: Routing distribution learnt by REINFORCE routing in the routing blocks of ResNet-DomainNet

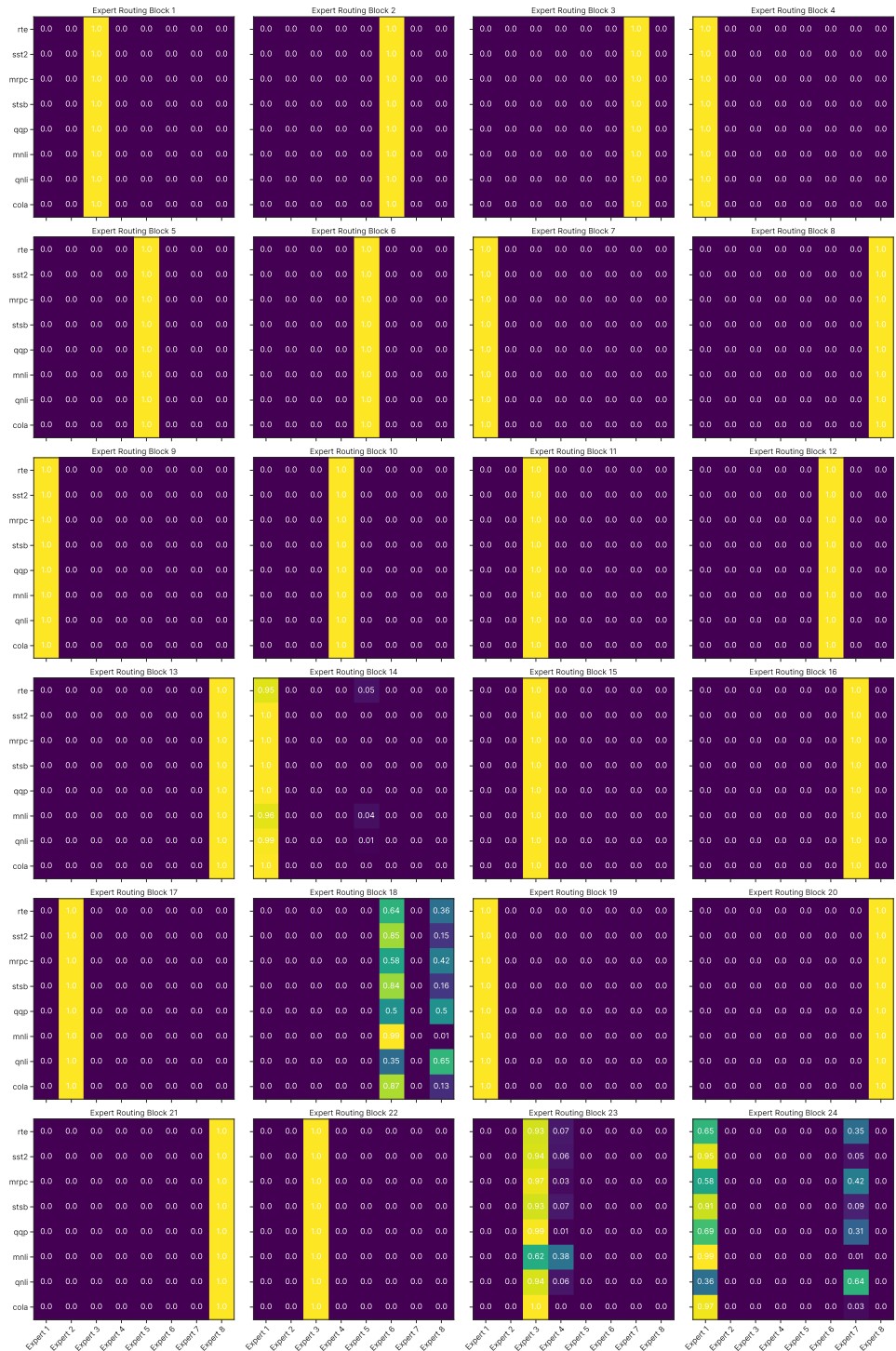

Figure 21: Routing distribution learnt by REINFORCE in the encoder routing blocks (1-24) of T5-GLUE

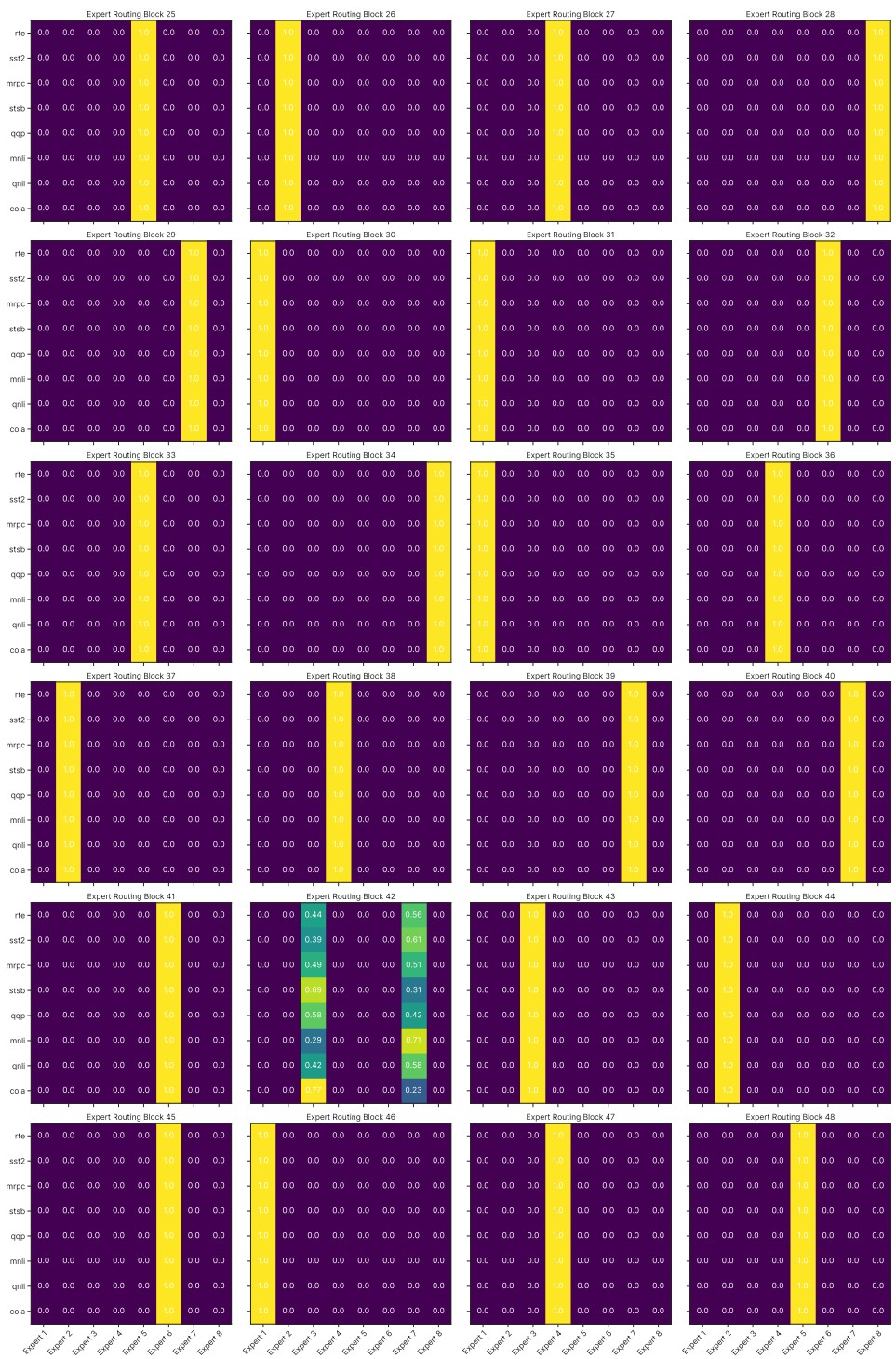

Figure 22: Routing distribution learnt by REINFORCE in the decoder routing blocks (25-48) of T5-GLUE

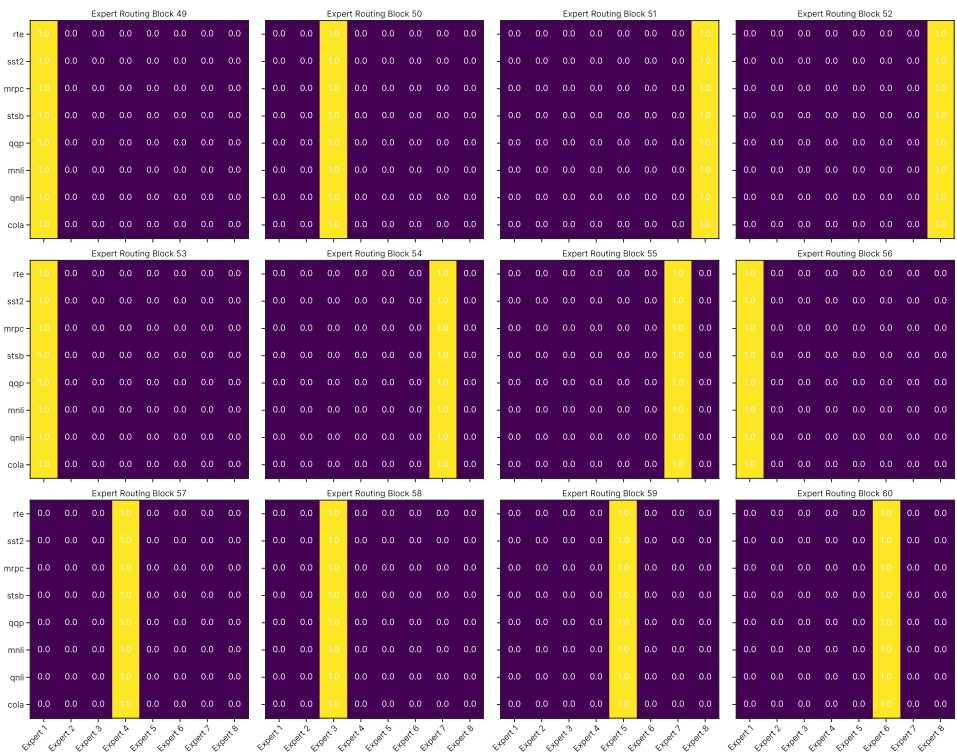

Figure 23: Routing distribution learnt by REINFORCE in the decoder routing blocks (49-60) of T5-GLUE

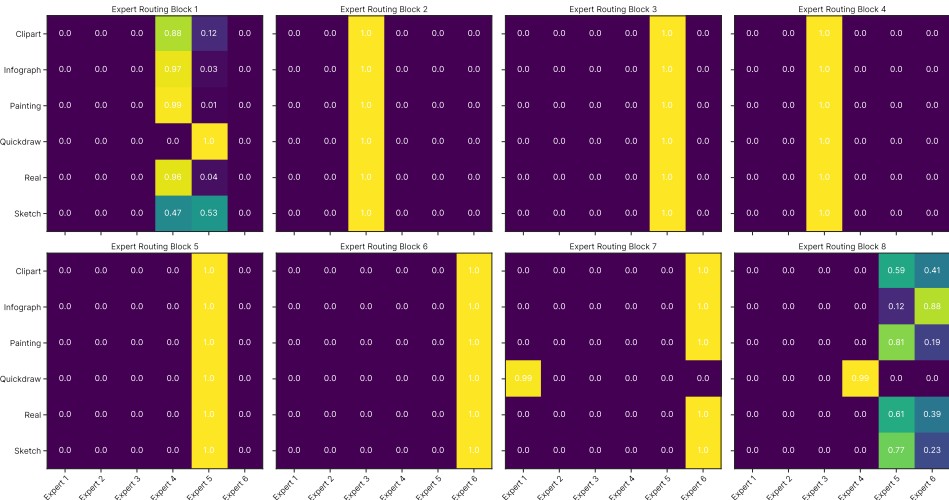

Figure 24: Routing distribution learnt by Dselect-$k$ routing in the routing blocks of ResNet-DomainNet

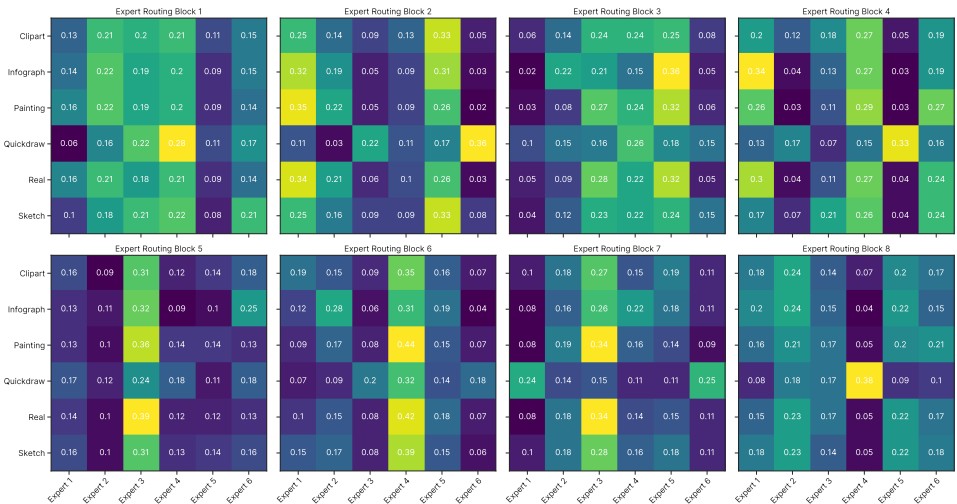

Figure 25: Routing distribution learnt by Ensemble routing in the routing blocks of ResNet-DomainNet

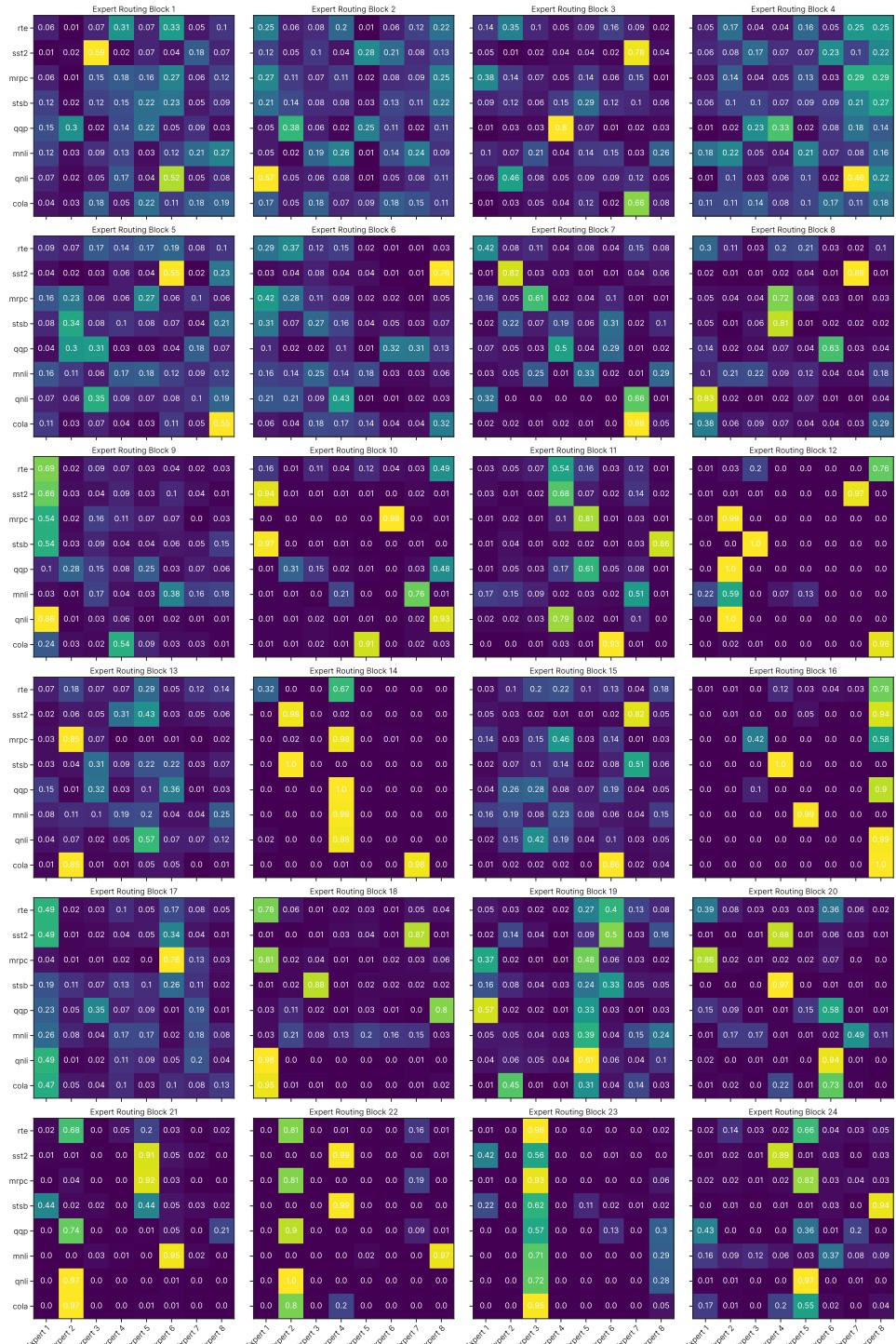

Figure 26: Routing distribution learnt by Ensemble in the encoder routing blocks (1-24) of T5-GLUE

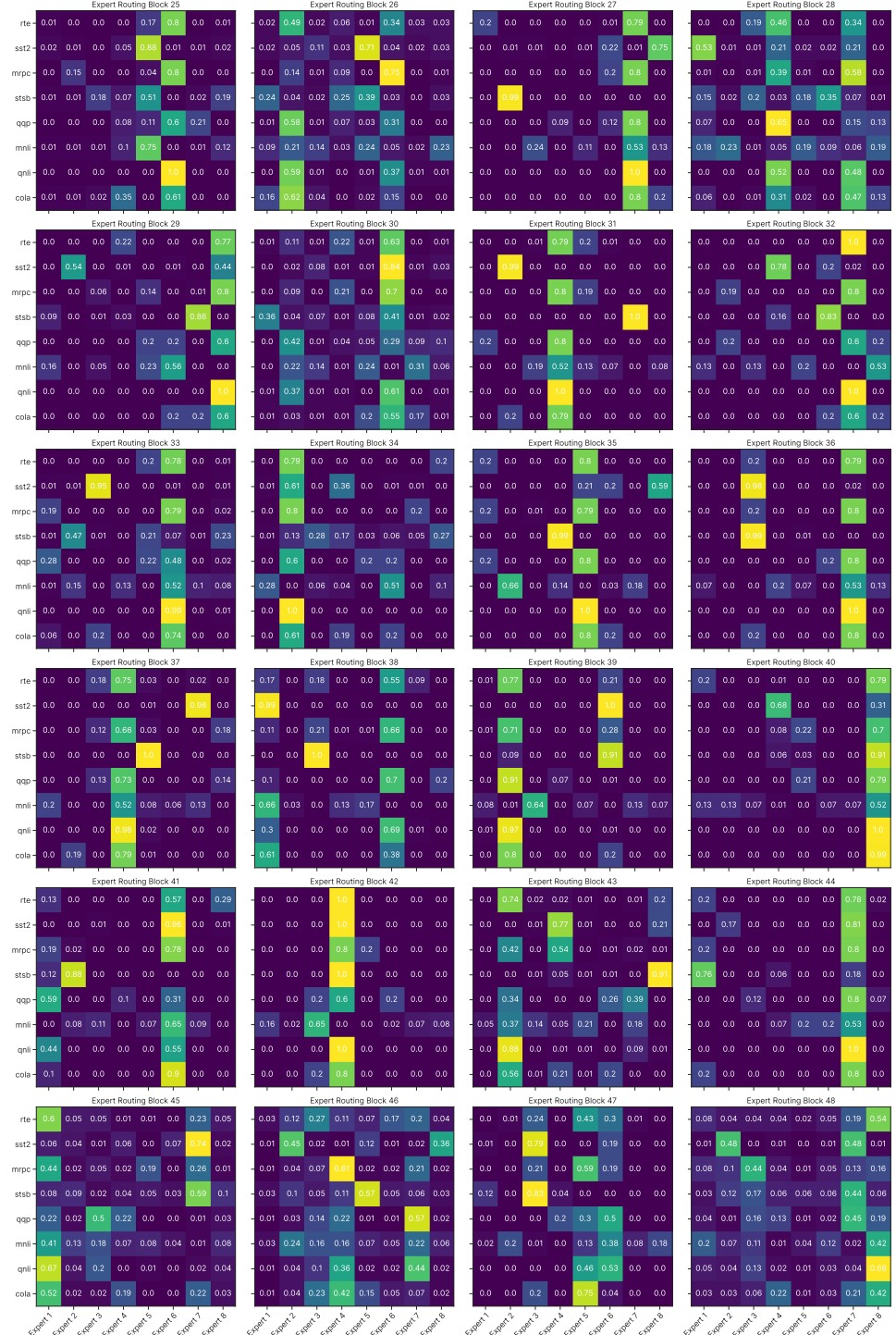

Figure 27: Routing distribution learnt by Ensemble in the decoder routing blocks (25-48) of T5-GLUE

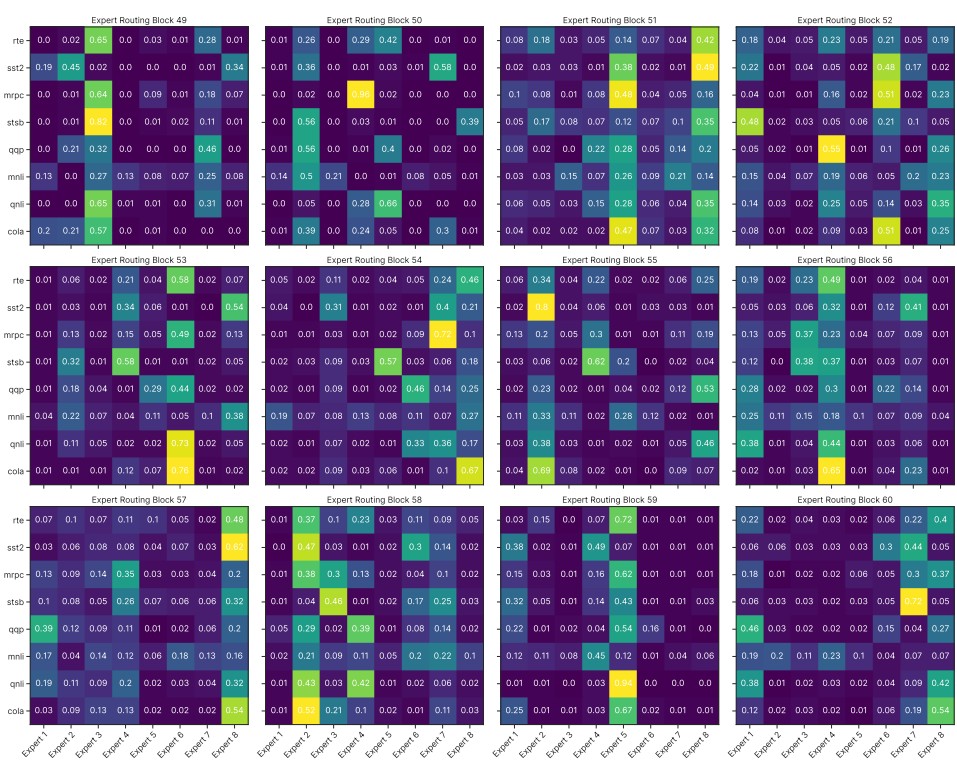

Figure 28: Routing distribution learnt by Ensemble in the decoder routing blocks (49-60) of T5-GLUE

