# OpenReview forum: "Soft Merging of Experts with Adaptive Routing"
_ICLR.cc/2024/Conference — Submitted to ICLR 2024_

### Official Review · Reviewer_y2t1 · 2023-10-31

**Soundness:** 2 fair
**Presentation:** 3 good
**Contribution:** 3 good
**Rating:** 6
**Confidence:** 3

**Summary:**

This paper proposes a simple method to avoid discrete routings in mixture-of-expert models by merging weights of experts during training and inference. Experiments over GLUE and DomainNet with T5-base and ResNet models demonstrate that the proposed approach is computationally efficient and achieves better performance than other MoE methods.

**Strengths:**

- The proposed method is simple, efficient, and achieves better performance than learned routing in presented experiments
- The choice of baselines are extensive, ranging from ensemble methods, learned routings, and pre-defined routings.
- I appreciate analysis of learned routing by the proposed method.

**Weaknesses:**

- The main weakness is the small scale of the data and models studied in the paper. I believe the challenge of reducing computational cost with mixture-of-expert models is more relevant to larger models. The authors however only presented results on ResNet and T5-base (with only 200M parameters). Experiment results with larger models are appreciated.

- If experiments with larger models are not feasible, I hope authors can discuss potential limitations of the study under those larger-scale scenarios. Do you expect the findings in Figure 2 change in larger scale setups?

- The author's hypothesis about the shortcomings of learned routings in existing works that "stem from the gradient estimation techniques used to train modular models that use non-differentiable discrete routing decisions" is not supported with evidence other than the performance of final models in Figure 2. If you visualize the learned routing of these baselines (as in Figure 3), will you notice flaws in these learned routing? Do they yield degenerated routing, or routing that is uniform across all the experts?

**Questions:**

Following listed items in weakness, I hope authors can address:
- What do you expect the performance looks like in larger-scale models?
- How does the routing learned by reported baselines look like?

Finally, the appendix and full text does not have to be in separate pdfs.

---

> ### Author Response · Authors · 2023-11-16
>
> We greatly appreciate the reviewers' thoughtful and detailed feedback. Here are our responses to the points raised:
>
> > The main weakness is the small scale of the data and models studied in the paper. I believe the challenge of reducing computational cost with mixture-of-expert models is more relevant to larger models. The authors however only presented results on ResNet and T5-base (with only 200M parameters). Experiment results with larger models are appreciated.
>
> While we agree that past MoE work has primarily focused on reducing computational costs of large models, our work instead focuses on the benefits of modularity in MoEs with the performance advantages they offer. This includes exploiting similar experts for examples from similar tasks to foster positive transfer and reduce negative interference among tasks. We specifically chose multitask learning as our benchmark because the MoE framework is intuitively appropriate for this setting. It's crucial to mention that each experiment, such as those with T5-GLUE, required approximately 100 hours on a single A6000 GPU, and the ResNet-DomainNet experiments took about 11 hours. This is because these benchmarks contain a set of datasets (eight in GLUE and six in DomainNet), making them quite extensive. Given these considerations, we opted for medium-sized models for our experiments. While we recognize the value of testing larger models, the current scope and our computational resources guided our choice of model scale.
>
> > If experiments with larger models are not feasible, I hope authors can discuss potential limitations of the study under those larger-scale scenarios. Do you expect the findings in Figure 2 change in larger scale setups?
>
> Our method is designed to be agnostic to the backbone architecture or model size. We incorporate MoE blocks into the existing architecture and focus on learning the router and experts within these blocks. Consequently, we anticipate that our method should effectively scale to larger-sized models. As for scaling the number of experts, as detailed in Section 4.1 of our paper, we have found that SMEAR does scale well. In the ResNet-DomainNet setting, it successfully narrows the gap to Ensemble routing. In contrast, in the T5-GLUE setting, the results show no significant difference, which might be attributed to performance saturation. If the reviewer has any specific large-scale experiment in mind, we would be willing to run them, resources permitting.
>
> > The author's hypothesis about the shortcomings of learned routings in existing works that "stem from the gradient estimation techniques used to train modular models that use non-differentiable discrete routing decisions" is not supported with evidence other than the performance of final models. If you visualize the learned routing of these baselines, will you notice flaws in these learned routing? Do they yield degenerated routing, or routing that is uniform across all the experts?
>
> Past work on gradient estimators such as  [https://arxiv.org/abs/1711.00123], [https://openreview.net/forum?id=r1lgTGL5DE, https://arxiv.org/abs/2109.11817] supports our hypothesis regarding the challenges associated with learning using gradient estimator techniques. In our paper, we adopt an empirical approach to compare learned routing methods, and therefore use final model performance as the metric for evaluating routing efficacy. Additionally, we have provided a qualitative analysis of learned routing methods in Appendix G. Here's a summary of their characteristics: In ResNet-DomainNet, Top-k demonstrates uniform routing in the initial layers but chooses a single expert in the last layer.  REINFORCE, ST-Gumbel, and Dselect-k tend to mostly exhibit degenerate single-expert routing policies. Interestingly, all these gradient estimators learn to assign a distinct expert for the Quickdraw dataset. However, this degree of specialization is insufficient for achieving superior performance scores. In T5-GLUE, these estimators display degenerate routing in some layers and a tendency to share a few experts (approximately 3 out of 8) in other layers across tasks. Methods like Latent Skills, Ensemble, and SMEAR, which utilize most experts in the MoE layer, generally outperform these estimators. It is important to note that Latent Skills requires task metadata, and Ensemble routing incurs significant additional computational costs. SMEAR not only performs better than Latent Skills but also approaches the effectiveness of Ensemble, all without needing task metadata and while maintaining a computational cost nearly equivalent to that of discrete routing methods. We incorporated this summary into the main text.
>
> > Finally, the appendix and full text does not have to be in separate pdfs.
>
> Thank you for your suggestion. We have now combined them into a single PDF for improved readability and accessibility.
>
> Thanks again for your time and please let us know if you have any further questions or clarifications.

---

> > ### Comment · Reviewer_y2t1 · 2023-11-21
> > **Response to authors**
> >
> > Thank you for your response. The authors improved the soundness of their results after revision. Please also include your discussion about learned mappings of baselines in the paper, either in main text or appendix.
> >
> > > Past work on gradient estimators such as [https://arxiv.org/abs/1711.00123], [https://openreview.net/forum?id=r1lgTGL5DE, https://arxiv.org/abs/2109.11817] supports our hypothesis regarding the challenges associated with learning using gradient estimator techniques. In our paper, we adopt an empirical approach to compare learned routing methods, and therefore use final model performance as the metric for evaluating routing efficacy.
> >
> > But the abstract says "In this paper, we hypothesize that these shortcomings stem from the gradient estimation techniques used to train modular models that use non-differentiable discrete routing decisions. " Do you consider the hypothesis to be one of the novel contributions? If the hypothesis is cited from prior works, the wording of the sentence should be updated to avoid the confusion. I think reviewer dSc4 raised a similar issue.

---

> > > ### Author Response · Authors · 2023-11-21
> > > **Re: Response to authors**
> > >
> > > Thanks for taking the time to read our response and providing additional feedback. We had previously added the details about routing patterns for baselines in section 4 under "Qualitative Analysis" in our updated draft.
> > >
> > > Regarding the sentence in the abstract, I think you're pointing out that we state a hypothesis in the abstract that we never explicitly prove. This is a good point, thanks for pointing it out. How about we change the abstract wording from "In this paper, we hypothesize that..." to "Past work has argued that..." - do you think that would make our scope clearer?
> > >
> > > Thanks in advance for any additional discussion.

---

### Official Review · Reviewer_uHR4 · 2023-11-01

**Soundness:** 3 good
**Presentation:** 2 fair
**Contribution:** 2 fair
**Rating:** 6
**Confidence:** 3

**Summary:**

This paper focuses on shortcomings of models with discrete routing among experts that can lead them to underperform heuristic non-learned routing. The authors hypothesize that issues with conditional computation stem from issues with gradient estimation, which is a technique utilized to provide approximate gradients for models involving discrete adaptive routing. The authors introduce SMERA for training modular models by computing a weighted average of the parameters of the individual experts.

**Strengths:**

1. Underperformance of models that use conditional computation is an important topic.
2. This paper provides some empirical analyses to verify the effectiveness of the proposed method.

**Weaknesses:**

1. The implementation details of the proposed method are not clear enough. For example, what routing function $R(\cdot)$ do you use? And what is the specific form of the objective function in the training process?
2. This paper lacks sufficient novelty. Can you explain what are the differences and advantages of the proposed method, compared to the $\pi$-Tuning method proposed in [1] (Section 2.2 and Section 3.5)?

[1] Wu, Chengyue, et al. "$\pi$-Tuning: Transferring Multimodal Foundation Models with Optimal Multi-task Interpolation." International Conference on Machine Learning. 2023.

**Questions:**

1. What routing function $R(\cdot)$ do you use?
2. What is the specific form of the objective function in the training process?
3. Can you explain what are the differences and advantages of the proposed method, compared to the $\pi$-Tuning method proposed in [1] (Section 2.2 and Section 3.5)?

[1] Wu, Chengyue, et al. "$\pi$-Tuning: Transferring Multimodal Foundation Models with Optimal Multi-task Interpolation." International Conference on Machine Learning. 2023.

---

> ### Author Response · Authors · 2023-11-16
>
> We thank you for your valuable comments and suggestions. Below we address each point raised:
>
> > The implementation details of the proposed method are not clear enough. For example, what routing function R(.)  do you use? And what is the specific form of the objective function in the training process?
>
> For the routing function R(.), we follow a standard approach of using a single linear layer which is a weight matrix of shape d x N, where d is the model dimension and N is the number of experts in the MoE layer. It takes in as input the average hidden states of tokens at the MoE layer and generates a probability distribution over N experts by applying softmax to the outputs. We included these details in our updated draft for greater clarity.
>
> Regarding training objectives, there are no additional loss functions added for the router. We use the language modeling loss for T5-GLUE and cross-entropy classification loss for ResNet-DomainNet and train expert and router parameters end-to-end with standard backpropagation. The benefit of our method being fully differentiable with these loss objectives is that it allows the router to be trained directly without any need for approximate gradient estimators or custom losses.
>
> > This paper lacks sufficient novelty. Can you explain what are the differences and advantages of the proposed method, compared to the $\pi$-Tuning method proposed in [1] (Section 2.2 and Section 3.5)?
>
> We want to emphasize that our method is simple, eliminating the need for gradient estimation and it outperforms previous routing methods. Moreover, our approach involves per-example adaptive routing without requiring any task meta-data, which is often difficult to obtain in real-world scenarios. In contrast,  $\pi$-Tuning employs a set of existing task specific experts, retrieving the top k experts for a downstream task and learns to interpolate among these experts for the downstream task. In our case, firstly, we do not assume access to a pool of task specific experts and secondly, we do not have access to task metadata. While  $\pi$-Tuning enables transfer learning to a new downstream task by learning to interpolate, our focus is on developing a routing algorithm among the combined dataset of tasks without distinct task boundaries and achieve positive transfer among existing tasks and reduce negative transfer which can be seen by the improved performance of our method. We added a more explicit discussion of $\pi$-Tuning in our updated draft.
>
> Thank you for your time. Please let us know if you have any further questions or clarifications.

---

> > ### Comment · Reviewer_uHR4 · 2023-11-22
> >
> > Thanks for your response.

---

### Official Review · Reviewer_dSc4 · 2023-11-02

**Soundness:** 3 good
**Presentation:** 4 excellent
**Contribution:** 3 good
**Rating:** 6
**Confidence:** 5

**Summary:**

This paper presents a novel method for routing experts in Mixture of Experts (MoE) models. Compared to methods that seek discrete routing policies and require gradient estimators (REINFORCE, Gumbel-Softmax), the authors propose an adaptive technique which is fully differentiable. Motivated by work on _merging models_,  SMEAR is presented, which computes a weighted average of expert parameters within each MoE block; thus enabling a fully differentiable routing policy. Authors validate on GLUE and DomainNet. They show that the method outperforms other discrete MoE models at equivalent inference rates.

**Strengths:**

1. This is very well written paper, clear it its aims, hypotheses, presented method and results. The manuscript offers an extensive analysis of the literature and corresponding baseline methods (gradient estimators, routing policies (top-k, d-select k etc.), heuristic routing, modular models) in addition to related work.

2. Merging expert parameters in MoE networks is under-explored and a straightforward *yet* elegant solution for adaptive routing of experts. The results are convincing although the improvement in accuracy does seem marginal at best (1% over REINFORCE Fig 2a, 2-3% TopK etc.)

3. The paper is a nice and interesting addition to the modular/MoE literature and presents interesting new research direction as pointed by the authors and beyond.

**Weaknesses:**

1. One weakness of the manuscript is the lack of a detail explanation on the central hypothesis of the work. The authors claim:

_"As we will later show in section 4, the gradient estimation techniques used to train models with discrete routing often fail to produce performant routing strategies."_

This is shown throughout quantitative results but as a reader, I was expecting a substantial theoretical explanation as to why discrete routing methods may fail related to how gradients may be routed, instabilities etc. Additionally, Figure 3 offers qualitative explanation of the learned policy through SMEAR. How does this relate to Top-K, DSelect-K etc. Are the policies that much different?

2. Many MoE methods often seen instabilities in training - mode collapse i.e. an expert being chosen much more, non-optimal solutions etc. The presented method is a weighted average of parameters. How does such a method regularise against particularly known failure modes of MoEs which often require explicit regularisation such as through load, importance, entropy, or Mutual Information?

**Questions:**

1. How would you go about encouraging exploration across experts in SMEAR to help determine which expert weighting is most beneficial per token/image etc.? In a discrete scenario, one might look at dropout and/or jitter noise on incoming representations? Did the authors try something similar to investigate performance?

2. The computational cost of the averaging works _"as long as there is a modest number of expert"_. In papers such as Switch Transformer, they use 128 experts compared to the 16 used in Scaling experiments. Would this method work in this context?

3. Can you foresee any pitfalls from where weight averaging might not be beneficial? What are the main limitations of the method?

4. How much specialisation do you learn as get one gets deeper into the network? Can you control this and would it get better results if you learned? There is a lot of learned sparsity in Figure3a but is this necessarily good? It could be a limitation that not enough exploration occurs through the proposed learning scheme.

[extra]
Formatting - not a question. but a comment on legibility. Section 4.1. needs to be broken up into paragraphs.

---

> ### Author Response · Authors · 2023-11-16
>
> Thank you for your insightful comments and constructive feedback. We have carefully considered your points and below is our detailed rebuttal:
>
> > The results are convincing although the improvement in accuracy does seem marginal at best (1% over REINFORCE Fig 2a, 2-3% TopK etc.)
>
> While the absolute improvements may appear small, we argue that they are meaningful since they are average values across a diverse range of datasets—8 in GLUE and 6 in DomainNet—conducted over 5 random seeds. In addition, improvements on these benchmarks are often small – e.g. most historically state-of-the-art methods on GLUE only improved over the prior state-of-the-art by a few tenths of a percentage. This not only underlies the robustness of our approach but also its adaptability across datasets and architectures. We added discussion to our paper on the significance of these gains.
>
> > I was expecting a substantial theoretical explanation as to why discrete routing methods may fail related to how gradients may be routed, instabilities etc
>
> We agree that theoretical analysis concerning the limitations of gradient estimators in discrete routing methods can be valuable. We chose not to include such analysis in our paper for two reasons: First, relevant discussions on this topic can be found in existing literature, notably in sources such as [https://arxiv.org/abs/1711.00123], [https://openreview.net/forum?id=r1lgTGL5DE, https://arxiv.org/abs/2109.11817]. Second, the theoretical advantage of our approach is simple: it facilitates the exact gradient computation via backpropagation. All past approaches for learning discrete routing require some form of gradient estimation, and these estimators incur bias or variance with respect to the true gradient. Our method has no such issues. We have edited our paper to more explicitly discuss this advantage and its relation to past theoretical analysis.
>
> >Additionally, Figure 3 offers qualitative explanation of the learned policy through SMEAR. How does this relate to Top-K, DSelect-K etc. Are the policies that much different?
>
> To answer your question, we have added routing policies for the various methods we considered in a new Appendix G. Here's a summary of their characteristics: In ResNet-DomainNet, Top-k demonstrates uniform routing in the initial layers but chooses a single expert in the last layer.  REINFORCE, ST-Gumbel, and Dselect-k tend to exhibit mostly degenerate single expert routing policies. Interestingly, all these gradient estimators learn to assign a distinct expert for the Quickdraw dataset. However, this degree of specialization is insufficient for achieving superior performance scores. In T5-GLUE, these estimators display degenerate routing in some layers, while showing a tendency to share a few experts (approximately 3 out of 8) in other layers across tasks.
>
> Methods like Latent Skills, Ensemble, and SMEAR, which utilize most experts in the MoE layer, generally outperform these estimators. However, it's important to note that Latent Skills requires task metadata, and Ensemble routing incurs significant additional computational costs. SMEAR not only performs better than Latent Skills but also approaches the effectiveness of Ensemble routing, all without needing task metadata and while maintaining a computational cost nearly equivalent to that of discrete routing methods. We incorporated this summary into the main text. We have not conducted a quantitative comparison of the routing policies across different methods due to the absence of established metrics, and such comparisons are nuanced. Therefore, we provide a qualitative analysis and use the final performance as a practical metric for evaluating the effectiveness of various routing methods.

---

> > ### Comment · Reviewer_dSc4 · 2023-11-23
> >
> > Thanks for your response. I’ve read the rebuttal across all reviews.
> >
> > My score will stay the same. Many thanks

---

> > > ### Comment · Area_Chair_9kw6 · 2023-11-23
> > >
> > > Thanks for the confirmation of your updated score.
> > >
> > > @authors, please leave your comments if necessary for the following AC-reviewer discussion.
> > >
> > > Best,
> > >
> > > AC

---

> ### Author Response · Authors · 2023-11-16
>
> > The presented method is a weighted average of parameters. How does such a method regularise against particularly known failure modes of MoEs which often require explicit regularisation such as through load, importance, entropy, or Mutual Information? How would you go about encouraging exploration across experts in SMEAR to help determine which expert weighting is most beneficial per token/image etc.? In a discrete scenario, one might look at dropout and/or jitter noise on incoming representations?
>
> Our method optimizes the routing policy using the loss function's gradient, focusing on minimizing loss. All past load balancing techniques are equally applicable to SMEAR, but they are inappropriate in the experimental context we consider because dataset sizes in the multitask mixture vary by multiple orders of magnitude and the load should therefore not necessarily be balanced. As concrete evidence, note that tag routing is an extremely unbalanced routing strategy but nevertheless has higher performance than most gradient estimators we considered. So, to promote exploration, we implemented expert dropout, randomly removing some experts in the MoE block and averaging the weights of the remaining ones. This strategy proved beneficial in specific contexts, notably in the T5-GLUE setting, as evidenced in our ablation study (Table 1). In contrast, we found that jitter noise did not offer additional benefits. In the case of other routing methods, we use expert dropout regularizer if it was found to improve performance (Table 1).
>
> >The computational cost of the averaging works "as long as there is a modest number of expert". In papers such as Switch Transformer, they use 128 experts compared to the 16 used in Scaling experiments. Would this method work in this context?
>
> Our reference to a “modest number of experts” is based on the computational ratio for discrete routing versus SMEAR, which is calculated as (1 + N x 2 / L x 4). Here, N represents the number of experts and L the number of tokens per example. As long as the second term of this ratio remains less than 1, the computational requirements are close. In our experiments with T5-GLUE, where L is 128 and N is approximately 8, the runtime difference was minimal. Moreover, tensor averaging typically benefits from GPU optimization. Therefore, we believe scaling up to N ~ 100 is feasible if the sequence length is around 1000, a scenario similar to what is observed in Switch models. This approach should maintain a computational cost comparable to discrete routing. We added this example to the paper for greater clarity.
>
>
> >Can you foresee any pitfalls from where weight averaging might not be beneficial? What are the main limitations of the method?
>
> While we recognize that SMEAR shows improved performance compared to other routing methods, a performance gap w.r.t. Ensemble routing still exists in the ResNet-DomainNet setting. Additionally, our approach necessitates that the experts be homogeneous, since the method of weight averaging works only in this case. Exploring better merging techniques to resolve this gap and allow for heterogeneous expert architectures would be exciting directions for future research.
>
> >How much specialisation do you learn as get one gets deeper into the network? Can you control this and would it get better results if you learned? There is a lot of learned sparsity in Figure3a but is this necessarily good? It could be a limitation that not enough exploration occurs through the proposed learning scheme.
>
> While we did not compute a quantitative measure of specialization as one progresses through the network's layers, our qualitative observations offer some insights. In the ResNet-DomainNet setting, we observed no significant difference in specialization across layers. Similarly, in the T5-GLUE setting, we did not find any obvious patterns in terms of layer-specific specialization.
> Regarding sparsity, a possible advantage is that it allows experts to specialize to certain tasks, but it is not obvious a priori that sparsity is valuable. Our method does not explicitly constrain sparsity and therefore can trade-off sparsity as appropriate. In the T5-GLUE experiments, we observed considerable sparsity, while in ResNet-DomainNet, despite the presence of specialization, sparsity was less pronounced. We believe exploring how to effectively regulate this specialization could be a promising direction for future research, potentially leading to improved routing algorithms.
>
> > Formatting - not a question. but a comment on legibility. Section 4.1. needs to be broken up into paragraphs.
>
> Thanks for the comment. We have updated the draft accordingly.
>
> Thanks again for your time. Please let us know if you have any further questions or clarifications.

---

> ### Comment · Area_Chair_9kw6 · 2023-11-22
> **[Time Sensitive, ICLR24] Please read the authors' responses and try to discuss the remaining concerns with the authors**
>
> Dear Reviewer dSc4,
>
> The authors have provided detailed responses to your comments.
>
> Could you have a look and try to discuss the remaining concerns with the authors? The reviewer-author discussion will end in one day.
>
> We do hope the reviewer-author discussion can be effective in clarifying unnecessary misunderstandings between reviewers and the authors, especially for this paper whose score is near the borderline.
>
> Best regards,
>
> Your AC

---

### Author Response · Authors · 2023-11-17
**Summary of Revisions and Responses to Reviewer Feedback**

Thank you everyone for your time and valuable feedback. Below is a summary of the key changes we've made, based on your insights.

- Highlighted that the reported improvements are meaningful, as they are averaged across a variety of datasets and multiple seeds.
- Discussed the theoretical advantage of our SMEAR method over gradient estimation techniques, emphasizing its end-to-end differentiability and exact gradient computation.
- Included a summary of our qualitative analysis comparing the routing policies of various learned routing methods.
- Expanded on exploration techniques, particularly detailing the use of expert dropout in SMEAR, which has shown benefits in specific contexts like T5-GLUE.
- Discussed the computational feasibility of SMEAR, especially when handling a large number of experts, and its comparision with discrete routing.
- Acknowledged SMEAR's limitations, particularly its performance gap compared to Ensemble in the ResNet-DomainNet setting, and the need for homogeneous experts for weight averaging.
- Provided insights into how SMEAR manages sparsity in its routing policy, demonstrating its ability to appropriately balance sparsity as needed.
- Clarified the specific training objectives and router architecture used in our experiments.
- Delineated our approach from the $\pi$-Tuning method, noting our focus on a routing algorithm that efficiently shares or specializes experts without the need for task metadata.
- Justified our choice of medium-sized models, considering the extensive benchmarks, baselines, and constraints on resources.

We have updated our draft accordingly and look forward to further discussion and feedback.

---

### Comment · Area_Chair_9kw6 · 2023-11-21
**[Time Sensitive, ICLR24] Please read the authors' responses and try to discuss the remaining concerns with the authors**

Dear Reviewers,

The authors have provided detailed responses to your comments.

Could you have a look and try to discuss the remaining concerns with the authors? The reviewer-author discussion will end in two days.

We do hope the reviewer-author discussion can be effective in clarifying unnecessary misunderstandings between reviewers and the authors.

Best regards,

Your AC

---

### Meta-Review · Area_Chair_9kw6 · 2023-12-06

**Metareview:**

In this study, the authors introduce an approach to expert routing in Mixture of Experts (MoE) models, offering a distinct alternative to existing methods that rely on discrete routing policies and gradient estimators like REINFORCE and Gumbel-Softmax. Their method, an adaptive and fully differentiable technique, is inspired by research on model merging. The proposed method, named SMEAR, calculates a weighted average of expert parameters within each MoE block, facilitating a fully differentiable routing policy. This approach is empirically validated on benchmark datasets such as GLUE and DomainNet. The results demonstrate that this method surpasses the performance of other discrete MoE models while maintaining equivalent inference rates.

The overall rating is borderline, and reviewers raised a lot issues regarding this paper. After the rebuttal, some of concerns are addressed, such as better presentation of this paper and model size discussion. However, no reviewer agrees that this paper should be accepted and still thinks that this is a borderline paper.

After carefully check this paper and all reviews, it can be found that some key concerns still remain in the revision. One reviewer indeed wants to know the theoretical support behind this paper, however, this concern is not fully addressed in the rebuttal. Besides, one major concern is about the model size used in this paper. One reviewer finds that the current experiments can rarely support the claim (not firmly). These concerns still exist, causing this paper to be a borderline paper even after the author-reviewer discussion.

Thus, in the consideration that 1) this paper is a borderline paper and 2) the novelty is not excellent, I recommend rejecting this paper at the current stage. We encourage the authors to fully address concerns from reviewers and submit a revision to the next top conference.

**Justification For Why Not Higher Score:**

After carefully check this paper and all reviews, it can be found that some key concerns still remain in the revision. One reviewer indeed wants to know the theoretical support behind this paper, however, this concern is not fully addressed in the rebuttal. Besides, one major concern is about the model size used in this paper. One reviewer finds that the current experiments can rarely support the claim (not firmly). These concerns still exist, causing this paper to be a borderline paper even after the author-reviewer discussion.

Thus, in the consideration that 1) this paper is a borderline paper and 2) the novelty is not excellent, I recommend rejecting this paper at the current stage. We encourage the authors to fully address concerns from reviewers and submit a revision to the next top conference.

**Justification For Why Not Lower Score:**

N/A

---

### Decision · Program_Chairs · 2024-01-16

Reject